# A comprehensive genetic map of cytokine responses in Lyme borreliosis

Javier Botey-Bataller [1,2,3,12], Hedwig D. Vrijmoeth [1,4,12], Jeanine Ursinus[4,5], Bart-Jan Kullberg[1], Cees C. van den Wijngaard[4], Hadewych ter Hofstede[1], Ahmed Alaswad[2,3], Manoj K. Gupta[2,3,6], Lennart M. Roesner [6,7], Jochen Huehn [6,8], Thomas Werfel[6,7], Thomas F. Schulz [6,9], Cheng-Jian Xu[1,2,3], Mihai G. Netea [1,10], Joppe W. Hovius[5], Leo A. B. Joosten [1,11,13] & Yang Li [1,2,3,6,13] ✉

The incidence of Lyme borreliosis has risen, accompanied by persistent symptoms. The innate immune system and related cytokines are crucial in the host response and symptom development. We characterized cytokine production capacity before and after antibiotic treatment in 1,060 Lyme borreliosis patients. We observed a negative correlation between antibody production and IL-10 responses, as well as increased IL-1Ra responses in patients with disseminated disease. Genome-wide mapping the cytokine production allowed us to identify 34 cytokine quantitative trait loci (cQTLs), with 31 novel ones. We pinpointed the causal variant at the *TLR1-6-10* locus and validated the regulation of IL-1Ra responses at transcritpome level using an independent cohort. We found that cQTLs contribute to Lyme borreliosis susceptibility and are relevant to other immune-mediated diseases. Our findings improve the understanding of cytokine responses in Lyme borreliosis and provide a genetic map of immune function as an expanded resource.

Lyme borreliosis (LB) is caused by tick-borne bacteria of the *Borrelia burgdorferi* s.l. species *(Bb)* after a tick bite. The incidence of LB has increased in the past decades[1,2]. Although timely antibiotic treatment cures the infection in the majority of patients, a substantial minority report persistent systemic symptoms such as fatigue, cognitive impairment, and pain[3]. The host immune system plays a crucial role in LB progression[4-7], with cytokines acting as key regulators of the interplay between the innate and adaptive responses to *Bb* and being correlated to the outcome of LB[8]. Imbalanced cytokine production during LB can lead to insufficient bacterial clearance or exaggerated inflammation and disseminated symptoms. Cytokine production is, however, highly variable, and depends on genetic and environmental

[1]Department of Internal Medicine and Radboudumc Community for Infectious Diseases, Radboud university medical center, Nijmegen, the Netherlands. [2]Department of Computational Biology for Individualised Infection Medicine, Centre for Individualised Infection Medicine, a joint venture between the Hannover Medical School and the Helmholtz Centre for Infection Research, Hannover, Germany. [3]TWINCORE, Centre for Experimental and Clinical Infection Research, a joint venture between the Hannover Medical School and the Helmholtz Centre for Infection Research, Hannover, Germany. [4]National Institute for Public Health and Environment (RIVM), Center for Infectious Disease Control, Bilthoven, the Netherlands. [5]Department of Internal Medicine, Division of Infectious Diseases & Center for Experimental and Molecular Medicine, Amsterdam UMC, University of Amsterdam, Amsterdam, the Netherlands. [6]Cluster of Excellence RESIST (EXC 2155), Hannover Medical School, Hannover, Germany. [7]Department of Dermatology and Allergy, Hannover Medical School, Hannover, Germany. [8]Department of Experimental Immunology, Helmholtz Centre for Infection Research, Braunschweig, Germany. [9]Institute of Virology, Hannover Medical School, Hannover, Germany. [10]Department for Genomics and Immunoregulation, Life and Medical Sciences Institute (LIMES), University of Bonn, Bonn, Germany. [11]Department of Medical Genetics, Iuliu Hatieganu University of Medicine and Pharmacy, Cluj-Napoca, Romania. [12]These authors contributed equally: Javier Botey-Bataller, Hedwig D. Vrijmoeth. [13]These authors jointly supervised this work: Leo A. B. Joosten, Yang Li. ✉e-mail: Yang.Li@helmholtz-hzi.de

factors. Previous studies[9–11] profiled the cytokine responses upon ex vivo pathogen stimulation of healthy donors, identifying the host genetic effects on cytokine production capacity, referred to as cytokine quantitative trait loci (cQTL). Dysregulated cytokine production is known to influence various immune-mediated diseases, such as inflammatory bowel disease (IBD)[12], multiple sclerosis[13], and LB[14]. In the case of LB, genetic variants affecting host-pathogen recognition[15,16], autophagy, or cytokine production capacity[17] have been linked to its pathophysiology.

In order to examine the immunological and genetic determinants of LB susceptibility, as well as the persistence of symptoms, a prospective cohort of 1138 patients with physician-confirmed LB was further investigated[18,19] (Suppl. Table 1). Patients were previously monitored for a year, and detailed clinical parameters were collected at five time points. Whole-genome assessment of genetic variation was performed, and antibody concentrations and cytokine responses to various stimuli were measured in 1060 cohort participants. Laboratory assessments were performed both at the start of antibiotic treatment (before or within seven days after the start of treatment, further referred to as "before treatment") and six weeks later, after most patients had completed antibiotic treatment. To profile cytokine production, four interleukins, IL-6, IL-10, IL-1Ra, and IL-1β, were measured upon 24-h stimulation with inactivated pathogens. IL-6 and IL-1β are known to mediate the innate response to $Bb$[20]. IL-1Ra regulates IL-1 signaling and is known to be less induced by $Bb$ than its pro-inflammatory part, IL-1β[21,22]. IL-10 exerts anti-inflammatory effects that regulate the adaptive response to $Bb$[5,14]. We explored the diversity in cytokine responses in these patients before and after antibiotic treatment. This allowed us to highlight differences in the responses due to the treatment or the disease phenotype. Moreover, we expanded the list of genetic loci associated with cytokine response (cQTL), providing a useful resource for future research on immune-mediated diseases.

## Results

### Regulation of cytokine responses upon stimulation

Cytokine responses were measured upon ex vivo stimulation of PBMCs and whole blood samples with various stimuli at two time points (Fig. 1a). First, we investigated which of these different conditions drive the variation in cytokine profiling. By reducing the total number of variables to their first two principal components, we observed a linear separation corresponding to the different cytokines (Suppl. Fig. 1a). Similarly, there was a high positive correlation among all cytokines, and clustering the data by their correlation pattern revealed that the measured cytokines themselves were the primary source of variation (Fig. 1b). The clustering results showed consistency between the two time points, before and after antibiotic treatment, suggesting the robustness of identified patterns (Suppl. Fig. 1b).

We observed that the variation in the data is primarily explained by cytokines rather than the stimulus[23], which contrasts with the previous results part of the Human Functional Genomics Project (HFGP)[9]. This may be due to the higher diversity of stimuli included in the HFGP study. Generally, the choice of stimuli and measured cytokines influence the main source of variation. In the current study on LB, the stimuli included were mainly derived from the causative pathogen of the disease (Fig. 1a), resulting in higher variance attributed to the measured cytokines (Fig. 1b).

We assessed the variation in cytokine responses with respect to clinical parameters, the course of infection, and the disease phenotype. Firstly, we found cytokine responses to be associated with age and sex (FDR <0.05; Spearman's and Wilcoxon rank-sum test). PBMCs isolated from male participants responded with higher IL-1Ra and IL-1β production. Age was positively correlated with IL-1Ra responses in PBMCs (Suppl. Fig. 2a). Secondly, we observed changes in cytokine responses after six weeks. The production of all four cytokines upon

$Bb$ stimulation increased after treatment, with a more evident effect seen in whole blood stimulations (Fig. 1c; FDR <0.05; Wilcoxon rank-sum test). Moreover, patients who had already started antibiotic treatment at baseline had a decreased IL-1Ra production compared to those who had not (Suppl. Fig. 3a–c; FDR <0.05; Wilcoxon rank-sum test). Thirdly, patients with disseminated disease responded with higher IL-1Ra production than those with early localized disease (Fig. 1d; FDR <0.05; Wilcoxon rank-sum test), regardless of sex effect (Suppl. Fig. 2c). Finally, IL-10 responses were negatively correlated with C6 antibody (IgG/IgM) index levels at both time points (Fig. 1e; FDR <0.05; Spearman's). Altogether, although different stimuli could lead to variation in measured cytokines, cytokines themselves affected variation most. Cytokine responses were positively associated with male sex, older age, the 6-week time point, and disseminated LB manifestations, whereas antibody responses were negatively correlated with IL-10 production.

### Genome-wide and study-wide significant cQTLs

Next, we performed cytokine quantitative trait loci (cQTL) mapping. Cytokine measures were normalized and quality controlled, and cytokine-stimulation pairs were excluded for QTL mapping in case of a non-normal distribution after log-transformation (Suppl. Fig. 4). In total, 60 cytokine-stimulation pairs were mapped to 7,706,179 SNPs in 1060 patients. This resulted in a total of 29 genome-wide ($P < 5 \times 10^{-8}$) and five study-wide ($P < 1.55 \times 10^{-9}$, Sidak correction on the number of effective tests) significant loci. Among the genome-wide cQTLs, the $TLR1$-$6$-$10$[9], $IL6$, and $IL10$[24] loci were reported in previous studies on individuals of European ancestry. In total, we identified 31 novel cQTLs. Most of the observed cQTLs were specific to a single cytokine from multiple stimulations, while only two were shared by more than one cytokine (Fig. 2). This finding aligns with the observation that the specific cytokine measured was the primary driver of variation in our system (Fig. 1b), to which genetic factors might have contributed. The five study-wide cQTLs (Fig. 2b) included the $TLR1$-$6$-$10$, $SLC6A19$, $IL6$, $EFR3A$ and $INTS6$ loci. $EFR3A$ codes for a protein that activates PI4K[25] and regulates IL-1β responses in PBMCs (Fig. 3a, locus 19, and Suppl. Fig. 1d–f). Expanding the list to the genome-wide cQTL (Suppl. Table 2) revealed additional relevant loci: the $CFH$, cis-$IL10$, and $KL$ loci. $CFH$ codes for the complement factor helper, which is related to the complement system and, thereby, to the immune response to pathogens. $KL$ codes for the protein Klotho, which has been linked to aging[26] and is involved in fibroblast growth factor (FGF) signaling (Suppl. Fig. 5a–c). To validate the effect of cQTLs on $Bb$ responses, we measured the gene expression of $Bb$-stimulated PBMCs from eight healthy individuals. We therefore identified as $Bb$-responding genes those differentially expressed compared to control-stimulated PBMCs, which allowed us to prioritize the identified cQTL (Fig. 2b and Suppl. Table 2). We found that genes functionally linked to $Bb$ cQTLs (through eQTL) were enriched in $Bb$-responding genes (GSEA, $P < 0.05$, Suppl. Fig. 6a). In summary, genome-wide cQTL mapping revealed a strong genetic impact on cytokine production capacity upon stimulations, including 34 significant loci, of which 31 were novel cQTLs.

### cQTLs are consistent across stimulations and time points

Next, we examined the association of the identified cQTLs with the different measurements and found that patterns were shared across the different conditions (Fig. 3a). The fact that cQTL were not specific to one condition can be explained by the common pathways that inter-regulate the different cytokines and the observed correlation patterns (Fig. 1b). A closer examination of the time differences (Fig. 3b) revealed a common direction of the effect for all the cytokine, stimulation and SNP combinations studied. Interestingly, we noticed that the replication capacity was a function of the minor allele count of the SNP (Fig. 3b). Variants with a minor allele frequency below a certain

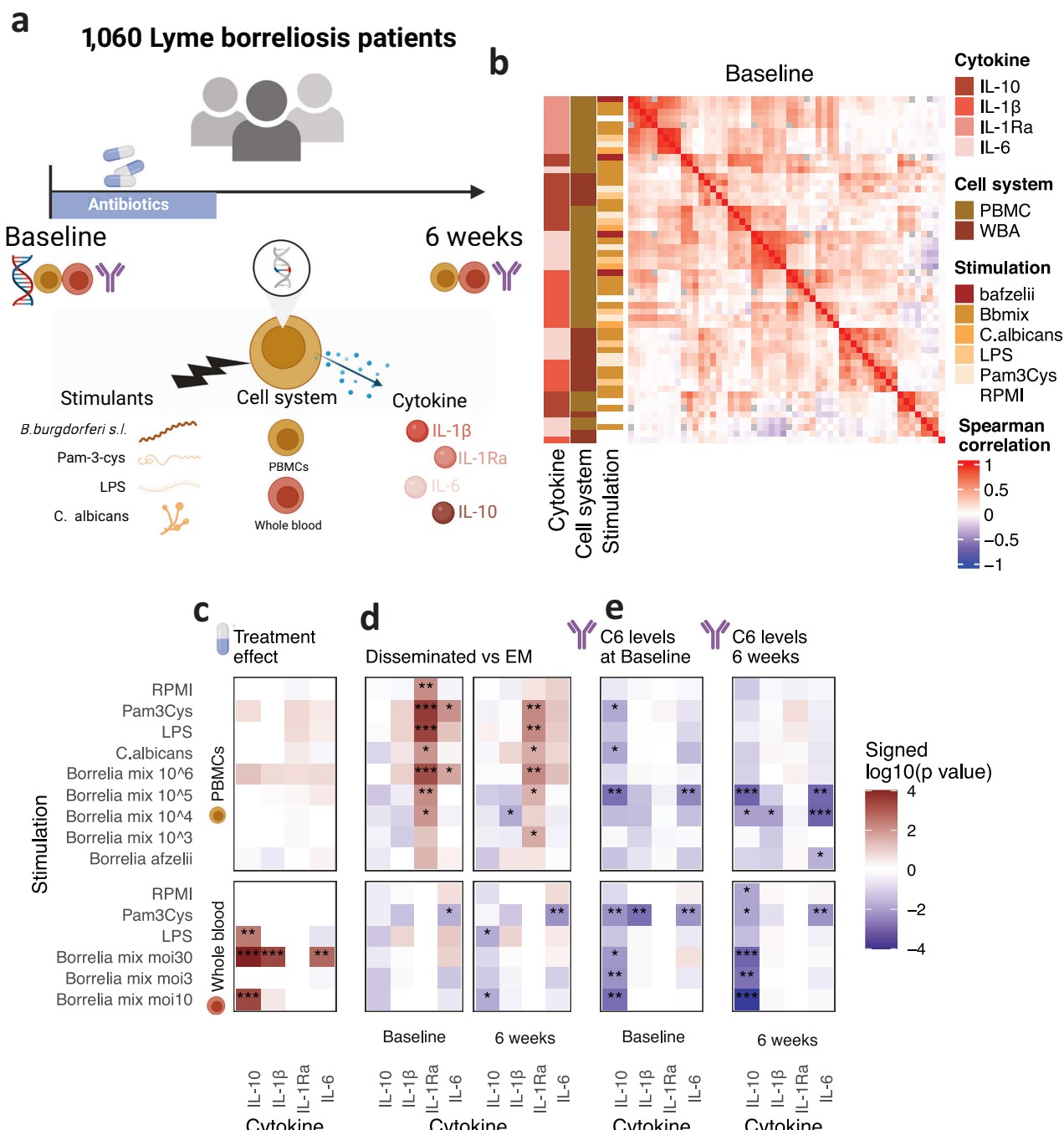

**Fig. 1 | Cytokine responses are organized around the different cytokines measured. a** Overview of the LymeProspect project. Schematic created with BioRender.com. **b** Heatmap of cytokine-stimulation correlation. Samples were taken at the start of antibiotic treatment. Spearman's correlation of the cytokine concentrations. **c–e** Heatmap of the significance of associations. *FDR <0.05, **FDR <0.01, ***FDR <0.001. Created with BioRender.com. **c** Paired two-sided Wilcoxon signed-rank test of cytokine values before and after antibiotic treatment. **d** Two-sided Wilcoxon rank-sum test of cytokine responses in patients with disseminated Lyme borreliosis and patients with erythema migrans. **e** Spearman's correlation of C6-ELISA, i.e., *B. burgdorferi* s.l antibody index., and cytokine responses.

threshold lacked the power to replicate the observed associations. Comparing the cQTL effects for two cytokines, IL-6 and IL-10, at the same conditions (Fig. 3c), we observed that two cQTLs had opposite effect directions, although the variants present in the loci had enough alternative alleles. This was due to the specific regulation of each cytokine through the *cis-* regulation of both gene sequences (Fig. 3d, e). We tested the colocalization of both *cis* effects on cytokine responses with *cis* effects on gene expression (*cis*-eQTL) from eQTLgen[27]. We observed a strong colocalization of the *cis*-IL10 locus (PP.H4 > 0.9), where the same sentinel variant was common to the

stimulation-cQTL and the eQTL effect (Fig. 3d), indicating that a change in transcription affects protein expression upon stimulation. The colocalization of stimulation *cis* cQTL to eQTL in the IL6 locus yielded more modest results (PP.H4 > 0.55) and two different sentinel variants for both summary statistics (Fig. 3e). This could indicate context-specific regulation, in which cQTL effects can only be detected in stimulation. Comparing our results with a recent QTL study on circulating proteins in the UK Biobank[28], we found a significant (FDR <0.05) association between Locus 8 (associated with IL-10 production upon LPS stimulation) and circulating IL-10 levels (Suppl. Fig. 7). This

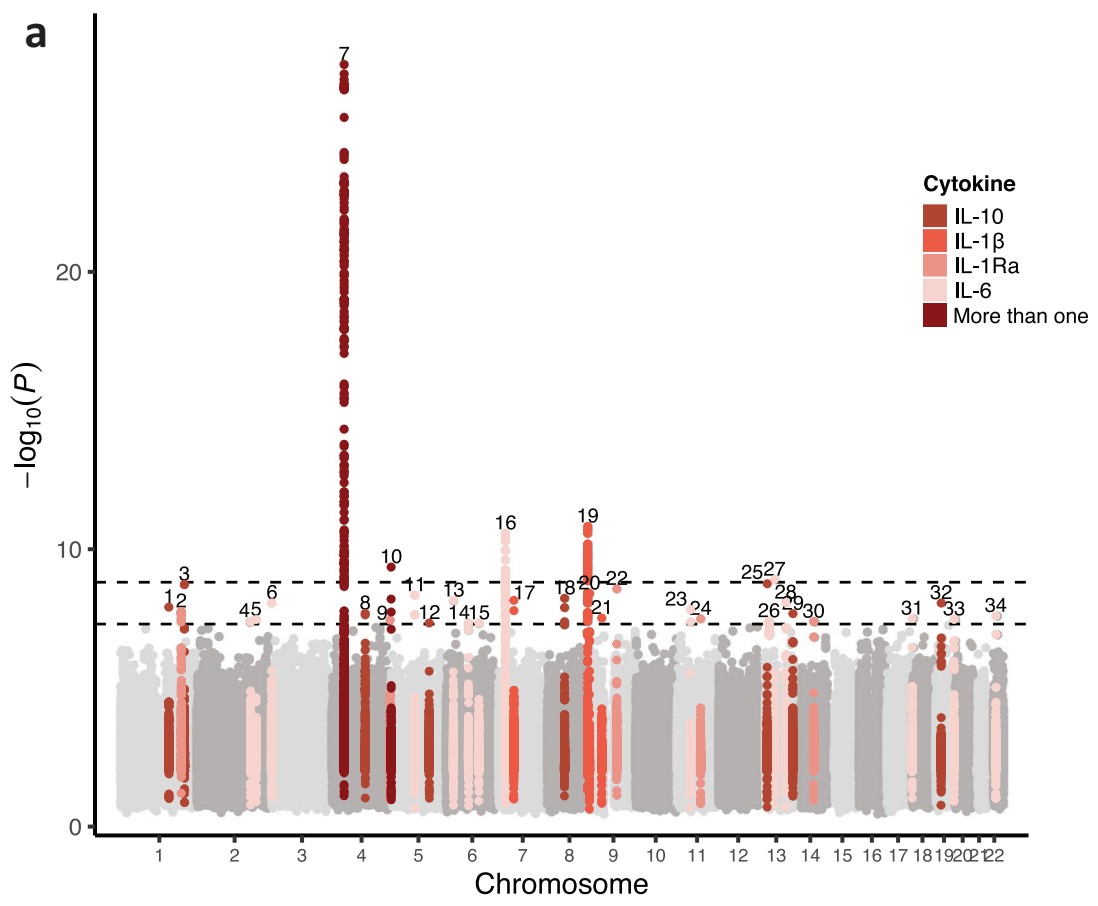

| Locus Nr. | Lead SNP | | | | | | | Cytokine | Stimulation | Time | Cell System | eQTL | Bb-responding |
| --- | --- | --- | --- | --- | --- | --- | --- | --- | --- | --- | --- | --- | --- |
| | SNP | Chromosome | Position | REF | ALT | ALT_AF | P | | | | | | |
| 7 | rs6815814 | 4 | 38816338 | A | C | 0.24198 | 3.35e-28 | IL-10, IL-6, IL-1b, IL-1ra | Bbmix 1e5, Bbmix MOI 3, Bbmix MOI 10, P3C, Bbmix MOI 30 | After treatment, Baseline | PBMC, whole blood | RP11-617D20.1, AC021860.1, TLR1, FLJ13197, TLR10, TLR6, KLF3, KLHL5, TMEM156, -, FAM114A1, NCS1, TAS2R31, TAS2R45, AKR1C2, LOC100653286, , CERS4, RPL9 | TLR1, TLR6, FAM114A1, RP11-617D20.1, AC021860.1, NCS1 |
| 10 | rs145818098 | 5 | 1197036 | A | G | 0.98969 | 4.37e-10 | IL-10, IL-6 | LPS, Bbmix MOI 30 | After treatment | whole blood | | |
| 16 | rs35345753 | 7 | 22740513 | C | G | 0.21178 | 2.65e-11 | IL-6 | C.albicans, Bbmix MOI 30, Bbmix MOI 10, LPS | After treatment, Baseline | PBMC, whole blood | TOMM7, IL6, AC005682.5, KLHL7-AS1, NUPL2, TARDBP, MASP2, LOC643387, AC006026.13, STEAP1B | IL6, STEAP1B |
| 19 | rs6990239 | 8 | 132831672 | C | T | 0.08891 | 1.49e-11 | IL-1b | Bbmix 1e5 | Baseline | PBMC | EFR3A | |
| 27 | rs76009888 | 13 | 51945061 | C | T | 0.97685 | 1.24e-09 | IL-6 | LPS | After treatment | whole blood | INTS6-AS1, RPS4XP16, INTS6 | |

**Fig. 2 | Genome-wide and study-wide significant cytokine QTL. a** Manhattan plot of cQTL. Colored loci indicate genome-wide significant cQTL. The colors indicate the cytokine for which the genome-wide association is found. **b** Table of study-wide significant loci ($P < 1.5 \times 10^{-9}$). Each locus is described by its top SNP and cytokine-stimulation for which it is found. The eQTL column indicates expression QTL effects, extracted from the eQTLgen consortium. Genes are labeled as "Bb-responding" if they are also responding to *B. burgdorferi*. All $p$ values were calculated using a linear model, associating cytokine measurements with genetic variants. Multiple-testing correction limits were set to genomewide significant ($P < 5e-8$), and study-wide significant ($P < 1.5 \times 10^{-9}$), based on the number of independent tests.

could reflect an SNP effect both in circulating and in functional production of IL-10.

In short, the observed cQTLs were shared across conditions, revealing a consistent effect after antibiotic treatment and specific regulation of the different cytokines and stimulations.

### Exploring the *TLR1-6-10* locus: fine mapping and differences in the regulation of pathogen responses

The *TLR1-6-10* locus yielded the strongest association among all the cQTLs identified (Fig. 2). Its effects were common for known TLR1-TLR2 ligands, such as *Bb* and Pam3Cys (P3C), and its signal reached significance in all measured cytokines at both time points and cell systems (Fig. 3a). The regulatory effect of this locus has been reported in previous studies applying TLR ligands in populations with European ancestry[9,23,24]. However, due to linkage disequilibrium (LD), it was challenging to dissect the causal SNP(s) in the locus, which could functionally impact any of the three TLR genes present and were not identified in previous cQTL studies. Fine mapping is a statistical method used in genetic studies to deconvolute the causal variant in association studies. Fine mapping enabled us to identify one non-synonymous variant, rs5743618-A, as the candidate causal variant (Fig. 4a, b and Suppl. Fig. 8a). rs5743618 was the candidate causal variant for all baseline cytokines and stimulations. However, it was not the top causal variant for measurements at the 6-week time point, as the results of fine mapping in that condition resembled more the results found in healthy cohorts[9] (Suppl. Fig. 8e–g). This resulted in different fine-mapped credible set (CS) sizes between both time points (Suppl. Fig. 8d). Further exploration revealed an increase in the effect sizes for most variants in the locus after treatment, while this change was close to zero for rs5743618-A (Suppl. Fig. 8c).

Adding to this, we compared the effect of the *TLR1-6-10* locus on cytokine responses to *Bb* stimulation and to P3C, as they are both ligands of the TLR1-2 dimer. We observed that the effect of rs5743618-A on IL-1Ra production upon *Bb* stimulation was significantly different upon P3C stimulation (Fig. 4c, d, Suppl. Fig. 9a–d, and Suppl. Data 3, $P_{\text{interaction-Baseline}} = 5.77 \times 10^{-7}$, $P_{\text{interaction-After treatment}} = 2.5 \times 10^{-6}$), indicating a differential genetic regulation between *Bb* and P3C stimulation. To validate these findings, we examined the genetic impact of rs5743618 on gene expression upon P3C stimulation in an independent cohort of 100 healthy individuals (TLR1 validation cohort, RESIST senior individuals (SI) cohort, see Methods). This analysis confirmed the association of the alternative allele (A) with higher IL-10, and a similar trend was observed for both IL-6 and IL-1β (Suppl. Data 4). We also found a negative association of the A allele with IL-1Ra expression response, confirming the differential regulation of IL-1Ra compared to the other examined cytokines upon P3C stimulation (Fig. 4e). Overall, we confirmed that rs5743618-A, a known *TLR1* missense variant, does not alter IL-1Ra production upon P3C stimulation, while it does affect all other cytokines and stimulations tested here (Fig. 4f).

In summary, the rs5743618-A was identified as the causal variant for the cQTL association of the *TLR1-6-10* locus and was found to differently regulate IL-1Ra responses to two TLR2 ligands: P3C and *Bb*.

### Identified cQTL are relevant for other immune-mediated diseases

We observed a high consistency of cQTLs between the samples taken at the start of antibiotic treatment and 6 weeks thereafter, when nearly all patients had completed antibiotic treatment. Moreover, previously identified cQTLs showed disease relevance[24]. Therefore, we hypothesized that the cQTLs identified in the cohort of patients with LB could be extrapolated to other infectious or non-infectious immune-mediated diseases. To test this hypothesis, we compared the aforementioned cQTLs to those found in the HFGP, which included 500 healthy individuals[9,10]. In that study, genetic variants with minor allele frequency (MAF) over 0.05 were used, whereas we used variants with MAF above 0.01. Therefore, we tested 9 cQTLs shared with the HFGP. We found 2/3 study-wide significant (loci 7, 16; Suppl. Figs. 10a) and 1/6 genome-wide significant (locus 3; Suppl. Fig. 10b) loci were significantly associated with cytokine responses in the HFGP (FDR <0.05). 5/6 genome-wide significant loci were significant before multiple-test correction ($P < 0.05$ and Suppl. Fig. 10b). Notably, the top variant in locus 3, which regulates IL-10 responses in LB, showed an opposite effect in IL-6 and IL-1β in the HFGP (Suppl. Fig. 10b), coinciding to what we observed in LB (Fig. 3a, locus 3). All in all, we found a moderate replication in the healthy of the cQTL found in LB[9,29].

Next, we tested the associations of immune-mediated diseases with the cQTLs, as found in public summary statistics from the GWAS catalog[30] (last accessed on 18/08/2022; Suppl. Data 5). In total, we observed significant (FDR <0.05) associations in four loci (Fig. 5a). In each of these associations, we tested the colocalization between our cQTL and the disease GWAS. Following this, we tested the causality of the associations applying single-variant Mendelian randomization (MR) on the top colocalized variant (Suppl. Data 6). First, we could link the *CFH* locus (locus 2), which we found associated with responses to *Candida albicans*, to age-related macular degeneration (AMD)[31] (Fig. 5b; single-SNP MR $P = 1.16 \times 10^{-5}$), in accordance with previous studies[32]. Second, we found a suggestively associated variant to ulcerative colitis (UC) and inflammatory bowel disease (IBD)[12] in the *cis*-IL10 locus, which yielded a significant colocalization and MR for rs3024493 (PP.H4 > 0.6 $MR_{\text{IBD}}$ $P = 1.16 \times 10^{-5}$, $MR_{\text{UC}}$ $P = 4.31 \times 10^{-6}$) (Fig. 5d). Previously, UC and IBD studies have mentioned the relevance of IL-10 in the pathology of the disease[33], specifically for its role in inflammatory responses to the host microbiome. Third, the *TLR1-6-10* locus has been associated with allergic disease[34] ($P < 1 \times 10^{-20}$). The sentinel variant for allergic disease[34] in that locus corresponded to the top fine-mapped variant in baseline conditions, which affects TLR1 trafficking to the membrane[35] (Figs. 5c, 4b), and was highly colocalized in the condition tested (PP.H4 > 0.9). Notably, higher cytokine production upon TLR2 stimulation was linked to a lower prevalence of allergy (Fig. 5f, MR $P = 3.48 \times 10^{-20}$). Fourth, locus 15 was suggestively associated with multiple sclerosis (MS)[36], and the sentinel variants of both associations were colocalized (rs79635669, PP.H4 > 0.9; MR $P = 8.24 \times 10^{-5}$) (Fig. 5e, f). The mechanisms of the CFH, TLR and cis-IL10 loci could be inferred with prior immunological knowledge and by their effects on closely situated protein-coding genes. The case of locus 15, however, was not evident. We observed that this locus was related to IL-6 responses in PBMCs upon all used stimulants and, weaker, to the other cytokines measured and whole blood IL-6 responses (Fig. 3a). IL-6 has been extensively studied in MS[13]. Taken together, we show that the cQTLs identified in LB could be validated in healthy individuals and are relevant for immune regulation and susceptibility to inflammatory diseases.

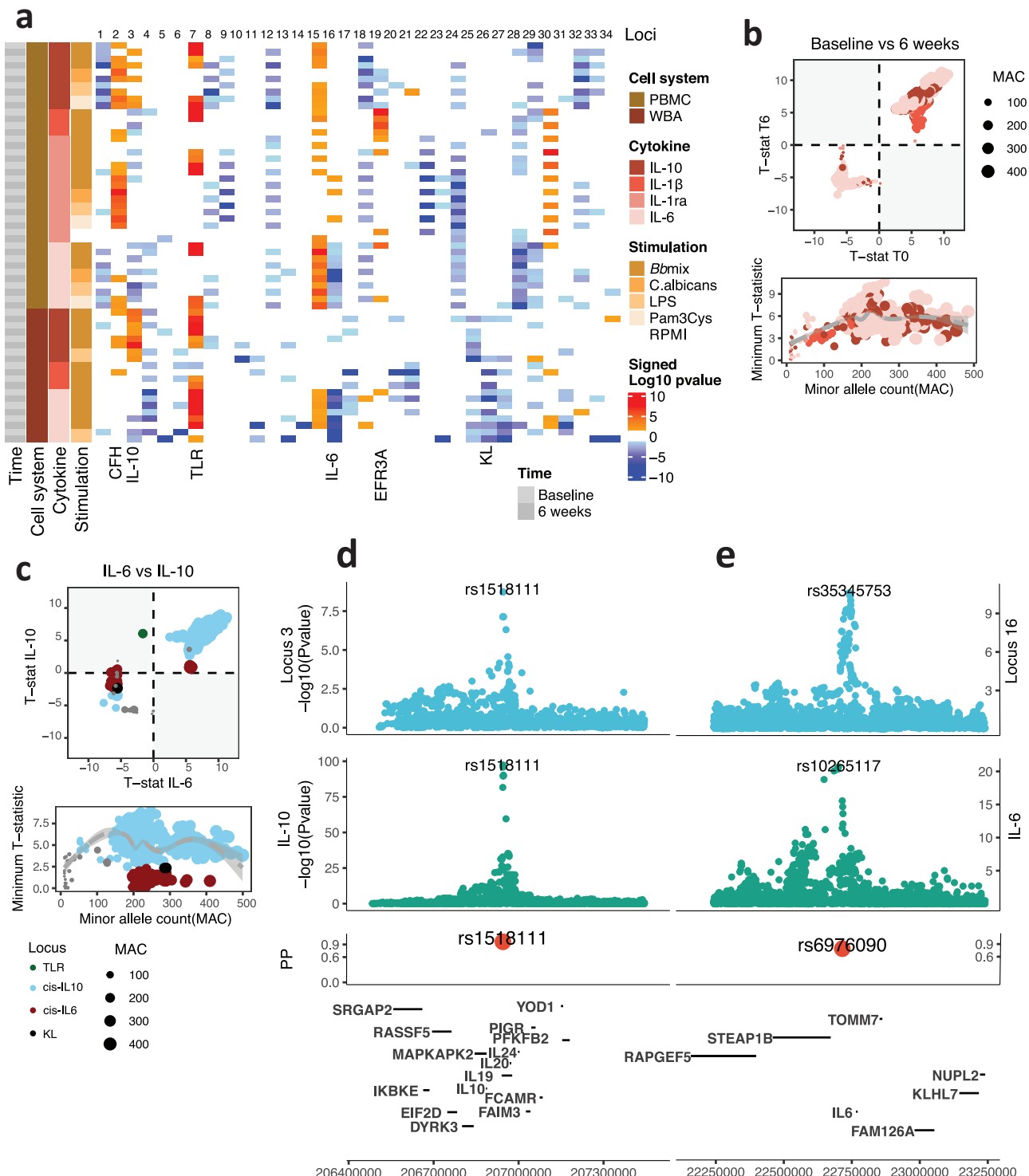

**Fig. 3 | Across condition consistency of genome-wide cQTL. a** Heatmap of cQTL signed $-\log_{10} p$ values for the different conditions studied. **b, c** Between conditions comparison. The top plot shows the comparison between t-statistics for both QTL mappings. The plot at the bottom shows the relationship between minor allele count and replicability, as shown by the minimum absolute t-statistic between the two conditions compared: both time points in (**b**) and IL-6 compared to IL-10 production in (**c**). **d, e** Two loci colocalized with blood eQTL signals. The Y-axis represents significance. The first plot is the locuszoom plot of cQTL. The second plot, locuszoom plot of blood eQTL. The third plot is the posterior probability of having a colocalized signal. $P$ values in (**a, d, e**), and t-statistics in (**b, c**) were calculated using a linear model, associating cytokine measurements to genetic variants. $P$ values are shown without multiple-testing correction.

## cQTL shed light on Lyme borreliosis susceptibility

Next, we tested whether the cQTLs we identified were associated with LB by integrating our data with a publicly available LB GWAS based on 342,499 individuals from the Finngen cohort[37,38]. In that study, a genome-wide significant locus (chr 11, *SCGB1D2*) was found to have a significant effect on bacterial clearance in the skin (Fig. 6a). Our data indicated that the risk allele was associated with an inflammatory response to *Bb* in whole blood, as well as an enhanced antibody production at both time points (Fig. 6b, c, Suppl. Fig. 11, and Suppl. Data 7, 8; coloc PP.H4 > 0.7). By connecting our data to the previous findings,

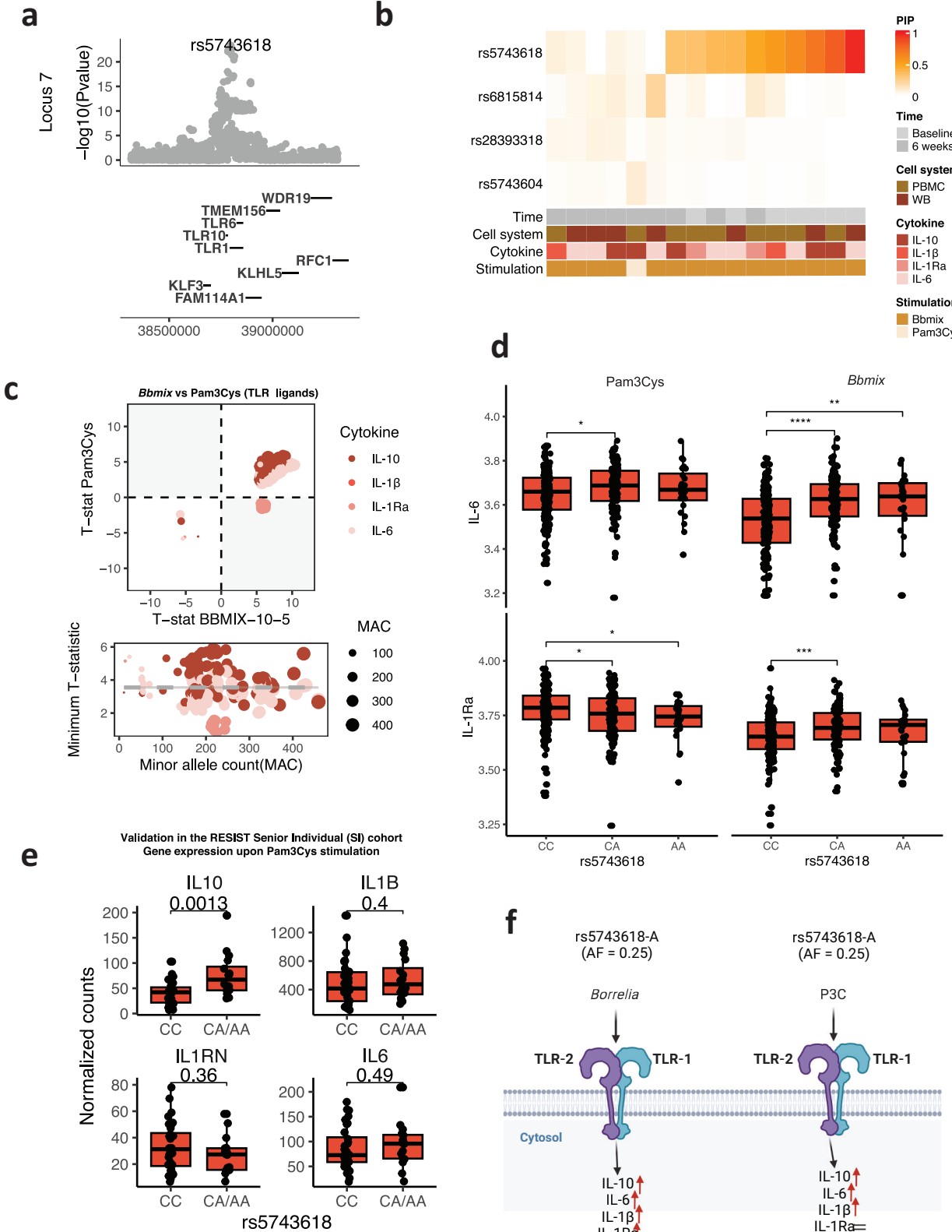

**Fig. 4 | Exploring the *TLR1-6-10* locus: Fine mapping and differences across conditions. a** Locuszoom plot of the top association found for the TLR locus. **b** Fine-mapping of the TLR locus per condition. The color indicates the probability of the variant being causal. PIP posterior inclusion probability. **c** Comparison of two TLR2 ligands. Same as Fig. 3b–d, **d** Highlighting the specific regulation of IL-1Ra responses to P3C. The Y-axis represents the log2 transformed cytokine concentrations; the X-axis, the different genotypes present for rs5743618 (*n* = 1060). **e** Replication of the specific regulation in the RESIST senior individual (SI) cohort

(*n* = 100). The Y-axis represents library-size corrected gene expression, X-axis, different genotypes present. **f** Schematic of the specific regulation of TLR1-2 responses. Created with BioRender.com. Boxplots in d and e represent the median value (middle line), 25th and 75th quantiles (box limits), and whiskers extend to 1.5 times the interquartile range. *P* values in (**a**), and t-statistics in (**c**) were calculated using a linear model, associating cytokine measurements to genetic variants. *P* values are shown without multiple-testing correction. *$P < 0.05$ **$P < 0.01$ ***$P < 0.001$.

we constructed a potential regulatory network of LB susceptibility. We hypothesized that individuals carrying the risk allele (rs4110197-G), when infected, would be less able to clear *Bb* from the skin, which would induce a stronger inflammatory response and increased antibody production (Fig. 6d). Finally, we used available data on spirochaetal infection susceptibility[38], which mostly concerned LB, to test the effect of cQTL on LB susceptibility. MR[39] analysis resulted in a suggestive causal link (unadjusted $P < 0.05$, multiplicative random effects IVW) between higher cytokine responses in two settings (IL-6 and IL-10 in *Bb*-stimulated PBMCs) and reduced susceptibility to LB (Suppl. Fig. 12 and Suppl. Data 9).

## Discussion

Here we report the characterized cytokine responses of 1060 Lyme borreliosis (LB) patients and the largest cytokine QTL mapping performed on any population to date. We found that cytokine responses upon stimulation of PBMCs and whole blood in these patients were highly variable and observed associations with sex, age, LB manifestation and antibody production. Moreover, we identified 34 significant cQTL, 31 of which were novel, with the *TLR1-6-10* locus showing the strongest association. Interestingly, this locus was associated with higher cytokine responses upon Bb stimulation, specifically IL-1Ra for *Bb*, but not for P3C. Furthermore, our cQTL findings may link to cytokine and immune responses in both health and disease. Finally, we were able to functionally link a locus that was previously described to affect LB susceptibility, SCGB1D2[37], to cQTL.

*Borrelia burgdorferi* s.l. *(Bb)*, the causative agent of LB, has been shown to disrupt innate cytokine responses[14]. In our study, we observed that ex vivo responses to *Bb* were enhanced at the 6-week time point. Previously, we reported that primary monocytes previously exposed to *Bb* exhibited decreased inflammatory cytokine production upon secondary stimulation[40]. This effect suggests the induction of innate immune tolerance, similar to the well-known tolerance effect in LPS conditions, which might explain the lower cytokine responses we here observed in patients at the time of active infection before or around the start of antibiotic treatment. Testing the possible immune-modulatory effect of the antibiotics, we found only IL-1Ra was diminished, suggesting that the lower response at baseline was not due to a direct antibiotic effect on the immune system. In addition, we found that a higher C6 (IgG/IgM) antibody index was associated with lower IL-10. This finding was consistent with studies performed in IL-10 deficient mice, which had higher antibody levels together with lower disease susceptibility[5,41,42]. Our findings indicate that the same mechanisms are true for humans, and one could speculate that modulating IL-10 production could be beneficial to mount a more effective antibody response. Furthermore, we found that patients with disseminated LB had higher IL-1Ra production compared to EM patients, particularly at baseline. This may relate to enhanced duration or severity of infection in disseminated manifestations, as IL-1Ra negatively regulates IL-1β to dampen inflammatory responses. Cerebrospinal fluid IL-1Ra concentrations showed a substantial decrease after antibiotic treatment in patients with Lyme neuroborreliosis in a previous study[43]. This could reflect the already-reported role of IL-1 signaling in disseminated LB, as IL-1β is detected at higher levels in patients with Lyme arthritis[22].

In recent years, several studies have performed cQTL mapping on different populations[9,10,23,24]. The present study includes the highest number of patients, with 60 cytokine-stimulation pairs (after quality control) at two time points. As a result, we were able to identify the highest number of loci and highlight new regulation of cytokine responses, as well as new interpretations in immune-mediated diseases. Initially, the HFGP[9] and other studies on individuals of European ancestry[24] portrayed the *TLR1-6-10* locus as the top cQTL. However, studies on individuals of African ancestry did not identify any regulation of cytokine production capacity at the *TLR1-6-10* locus[23]. Here we were able to fine-map this locus and highlight its causal variant and its context-dependent regulation. rs5743618-C, produces an alternative TLR1 structure in its transmembrane helices[44], thereby reducing TLR1 trafficking[35], which influences TLR1-TLR2 activation, and leads to lower cytokine production, as observed in our data. Here, we found rs5743618 to be the single causal variant in cytokine responses before antibiotic treatment but not after antibiotic treatment, suggesting a context-specific cQTL effect that we could confirm from the HFGP data, where rs5743618 was not selected as the top variant. We compared the regulation of cytokine responses upon the two TLR2 ligands, P3C and *Bb*, by the *TLR1-6-10* locus. We found, and could validate in an independent cohort, that this locus did not regulate IL-1Ra concentrations upon P3C stimulation. This might be a consequence of the higher specificity of P3C, which is only a ligand for TLR2, compared to *Bb*, which is known to activate different pathogen recognition receptors.

We also identified new regulators of cytokine responses to pathogens. In cQTL, variation in a locus at the Klotho gene (*KL*) was associated with whole-blood inflammatory responses (IL-6 and IL-1β) to various stimulants. Previous studies have shown the anti-inflammatory[45] and aging[26] properties of Klotho, together with its role in responses to TLR ligands[46]. Specifically, Klotho has been associated with the two cytokines that its genomic locus is regulating: IL-6[47] and IL-1β[48]. We report that *KL* was related to inflammatory responses to pathogens. This could provide a better understanding of the pleiotropic effects of *KL* and, therefore, its key role in aging. Moreover, we identified a new gene (*EFR3A*) that could be relevant for the response to pathogens. The *EFR3A* locus showed one of the strongest associations ($P < 2 \times 10^{-11}$) with IL-1β responses upon *Bb* stimulation of PBMCs. This locus has been linked to PI4K activation[25], a highly conserved complex that plays a role in lipid transport and autophagosome formation[49].

We found that the cQTLs identified in the HFGP are relevant for susceptibility to infection[9]. We confirmed that *Bb* cQTLs were associated with LB susceptibility and exemplified the use of the identified cQTLs in other immune-mediated diseases. Firstly, genetic protection against age-related macular degeneration (AMD) was linked to higher IL-1Ra and IL-10 responses to a wide range of pathogens through the *CFH* locus. *CFH* and the complement system had already been studied in the context of AMD[32]. IL-1Ra and IL-10 had also been studied in patients with AMD[50,51]. Secondly, genetic protection against allergy was related to higher cytokine responses to TLR2 ligands through the *TLR1-6-10* locus. As we pointed out before, the fine-mapped variant in the *TLR1-6-10* locus, which is known to alter the trafficking of TLR1 to the monocyte membrane[35], led to a different cytokine response to TLR1-TLR2 activation. Other studies have shown that the minor allele (rs5743618-A) increases the activation of effector T-cells[52]. Thirdly, genetic susceptibility to multiple sclerosis (MS) was linked to higher IL-6 responses through the *QRSL1* locus. Although IL-6 has previously been associated with MS activity and pathophysiology[13], the genes present in the *QRSL1* locus, have never been linked to immune-related pathways. QRSL1 is a subunit of the Glutaminyl-γRNA amidotransferase. It has been linked to mitochondrial dysfunction[53], but its role in inflammation or MS is unknown. Lastly, we found that ulcerative colitis (UC) and inflammatory bowel disease (IBD) were related to IL-10 responses through the *cis*-regulation of *IL10* locus. The role of IL-10 in IBD and UC has been widely studied[12].

Recently, a study conducted on the Finish population identified two loci that were associated with LB susceptibility[37,38]. The top variant was linked to a non-synonymous SNP in *SCGB1D2*, resulting in an alternative protein structure that inhibited bacterial growth less efficiently. To decipher the acting mechanisms of the variant on the host immune system, we used data from our prospective cohort. We found that individuals harboring the risk allele had a stronger induction of inflammation upon *Bb* stimulation in whole blood. Moreover, antibody

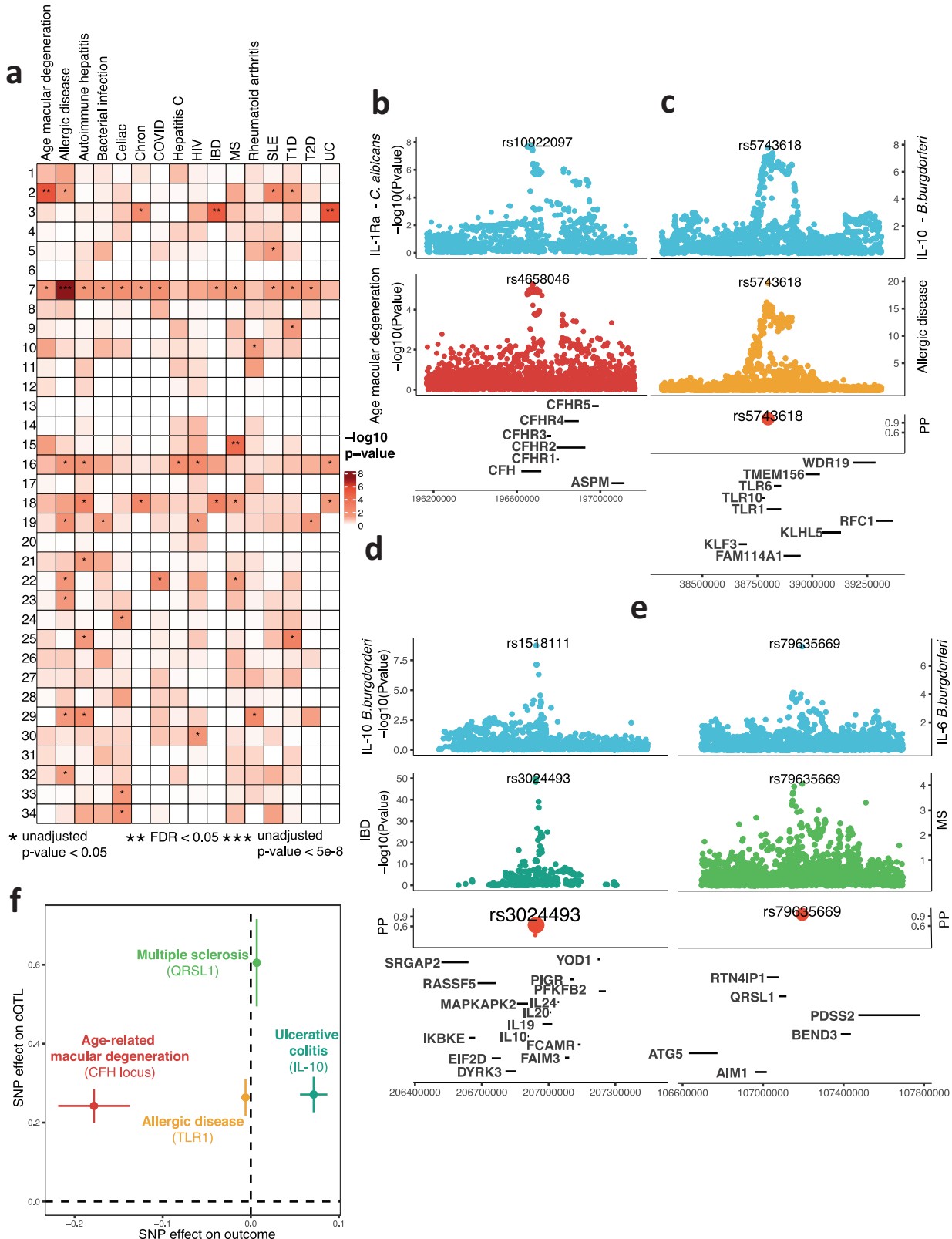

**Fig. 5 | The identified cQTL are relevant for different immune-mediated diseases. a** Heatmap indicating the significance of the cQTL (rows) in public GWAS results from immune-mediated diseases (IMD) (columns). **b–e** Colocalization of the significant associations (FDR <0.05) between cQTL and IMD GWAS. **f** Causal link between cQTL and IMD GWAS. All *p* values in (**a–e**) were calculated using a linear model, associating cytokine measurements to genetic variants.

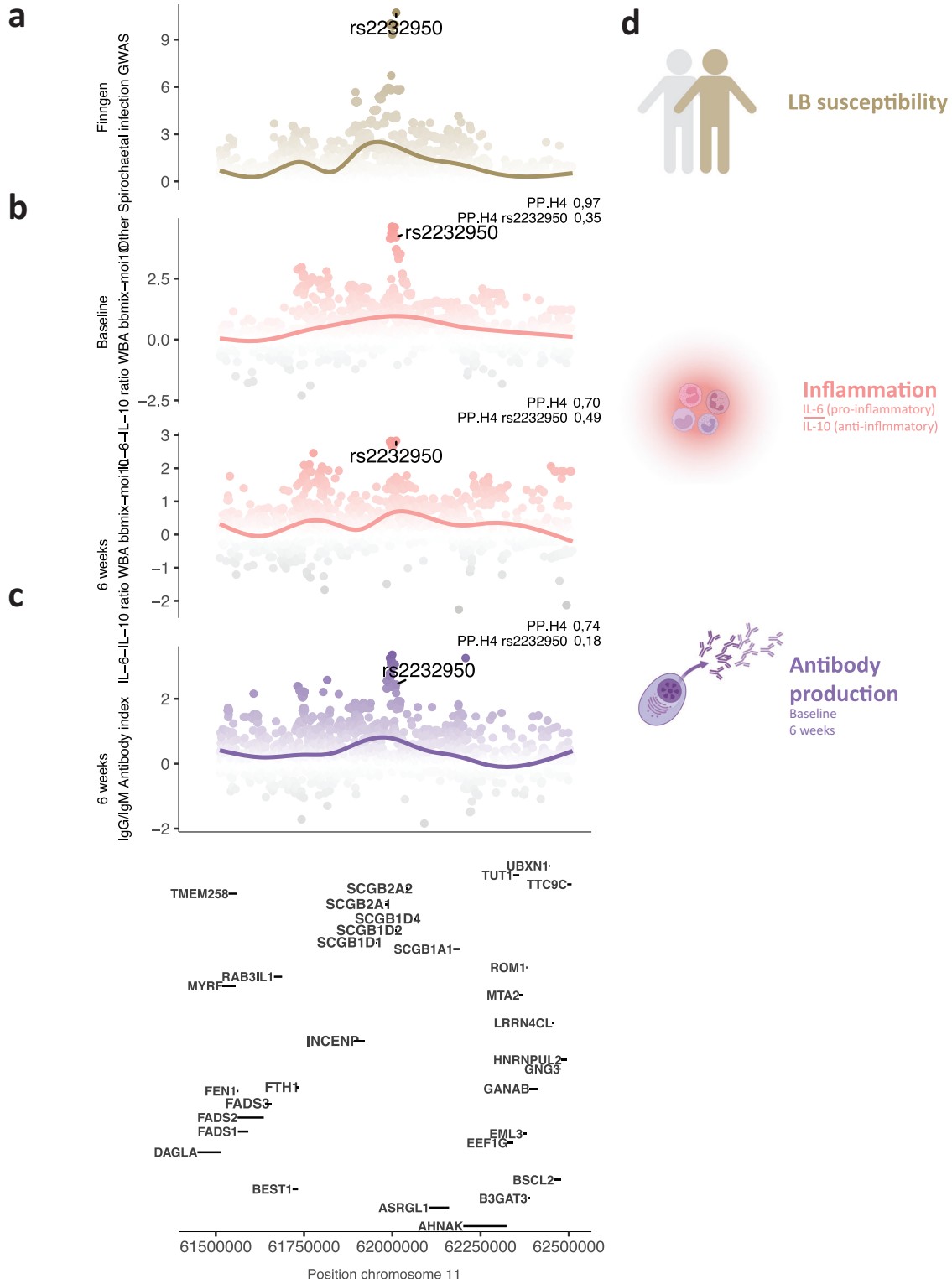

**Fig. 6 | cQTL shed light on Lyme borreliosis susceptibility. a** Lyme borreliosis susceptibility locus previously reported[37]. **b** Locuszoom plot of the same locus for the ratio between IL-6 and IL-10 production upon *bbmix* stimulation in whole blood levels were higher at both time points.

at baseline. Colored by inflammation induction (higher IL-6 to IL-10 ratio). **c** Same as **b**, at the 6-week time point. **d** Schematic of the proposed effect exerted by the Lyme borreliosis susceptibility locus. Created with BioRender.com.

These findings led us to hypothesize a regulatory network linking reduced bacterial clearance to more inflammation, which, in turn, activates the adaptive immune system. Consequently, increased inflammation could lead to more severe symptoms and higher antibody levels, which are likely to result in a positive serological test and earlier diagnosis of LB.

We tested the effect of cytokine production capacity on LB susceptibility by applying Mendelian Randomization methods. Our results showed a suggestive causal link between higher IL-6 and IL-10 production and protection against LB. However, the results should be interpreted with caution, as we used a suggestive significance threshold (nominal $P < 0.05$), and the sensitivity tests indicated a probable

case of pleiotropy ($P_{IL\text{-}6} = 0.32$, $P_{IL\text{-}10} = 0.06$; Suppl. Data 9) and excess in heterogeneity ($P_{IL\text{-}6} = 0.06$, $P_{IL\text{-}10} = 0.04$; Suppl. Data 9).

Collectively, we present a comprehensive profile of cytokine responses in LB, integrating these responses with the course and phenotype of the infection. We demonstrate the role of genetic variation in cytokine responses in LB, expanding on previous studies on healthy individuals and create a genetic map of the regulation of cytokine production capacity, which is made available via a web tool at https://lab-li.ciim-hannover.de/apps/lyme_cqtl/. Finally, we exemplify the use of cQTL to explain genetic susceptibility to infection and test the causal role of cytokine production.

## Methods

### Study design: LymeProspect cohort recruiting−clinical data

The LymeProspect cohort consisted of 1138 patients with a new physician-confirmed LB (either EM or disseminated manifestation) who were included at the initiation of antibiotic therapy and followed for one year, between April 2015 and October 2018[18,19]. At baseline and every three months during follow-up, clinical data were collected with online questionnaires. Blood samples were taken at baseline and 6 weeks thereafter. The medical ethics review committee Noord-Holland (NL50227.094.14) approved this study. It was conducted in accordance with the Declaration of Helsinki, and all patients gave written informed consent.

### PBMC/whole blood isolation and stimulation experiments

Part of this project focused on the cytokine responses to various stimulants and their genetic determinants in relation to antibiotic therapy (at the start or 6 weeks later). Therefore, ex vivo stimulation experiments were performed with isolated PBMCs and whole blood. PBMCs were isolated using a Ficoll density gradient within 24 h after blood collection. After isolation, cells (in a concentration of $5 \times 10^6$/mL RPMI) were added 1:1 with stimuli to the wells of a round-bottom 96-well cell culture plate. In 48-well cell culture plates, 100 µl of heparinized whole blood was stimulated with 400 µl of stimulus. Stimuli included live attenuated *B. burgdorferi* s.l. in a proportional mix of *B. burgdorferi* s.s. ATCC strain 35210, *B. afzelii* pKo isolate, and *B. garinii* ATCC strain 51383 (referred to as bbmix) or *B. afzelii* only, at different concentrations (i.e., $5 \times 10^3$ to $5 \times 10^6$ spirochetes per mL for PBMC stimulation, or MOI 3, 10, or 30 for whole blood stimulation). In addition, lipopolysaccharide 100 ng/ml (purified LPS from E. coli serotype 055:B5), Pam3Cys 10 µg/ml, and heat-killed *Candida albicans* blastoconidia $1 \times 10^6$/mL (for PBMC stimulation only) were used as stimuli (Fig. 1a). RPMI 1640 (Dutch modification) was used as the medium control. Cells were incubated for 24 h at 37 °C and 5% $CO_2$, after which supernatants were stored at −20 °C until further assessment[17].

Commercially available ELISA kits (Sanquin Reagents, Amsterdam, the Netherlands, and R&D Systems, Minneapolis, MN, USA) were used to measure interleukin (IL)−1β, IL-6, IL-10, and IL-1Ra in the supernatants, according to the instructions applied by the manufacturer.

### DNA isolation and genotyping

DNA was isolated from EDTA whole blood using Qiagen's DNeasy® Blood & Tissue Kit (Hilden, Germany), in accordance with the accompanying protocol. For genotyping, the Infinium Global Screening Array MD v1.0 (GSA) BeadChip (Illumina, San Diego, CA, USA) was used.

### Genotype quality control and imputation

The reported sex from the donor was checked with the genotyped sex, discarding samples with discordances. A full summary of the samples discarded can be found in Suppl. Fig. 13. Eight donors for which more than 3% of the SNPs had missing information were discarded. Twenty-three donors with a heterozygosity rate more extreme than three standard deviations from the mean were also discarded. The relatedness was checked; six samples exhibited a high degree of relatedness, and the sample with higher missingness of each pair was discarded. Donor ancestry was compared to the known ancestries of the 1000 genomes project[54], discarding 23 donors that were too distant to European ancestry. After these quality control filters, the genotype was imputed using the Michigan imputation server[55]. Samples were imputed using HRC 1.1 2016 (hg19)[56] as the reference, using Eagle v2.4[57] for phasing, matching to a population with European ancestry, and applying no filtering on the r squared value.

After imputation, variant information was loaded to plink2[58] using the imputed dosage by minimac4. SNPs were again filtered, an r-squared value below 0.5, a minor allele frequency below 0.01, or a p value for Hardy−Weinberg equilibrium below 1e-12 was used to discard the variant. After filtering, 7689871 SNPs from 1060 LB patients were included for further analyses.

### QTL mapping

Cytokines used for trait mapping were also filtered. Values were transformed by applying the base 2 logarithm. Stimulation-Cytokine pairs with non-symmetric distribution were not used for QTL mapping. Cytokine QTL mapping was performed using a linear model from the R package MatrixEQTL[59], including the covariates batch (the two genotype batches), the institute where stimulation experiments were performed (Radboudumc or Amsterdam UMC), age and sex. The resulting stimulation-cytokine pairs were examined based on the p value and the beta value returned by the model.

### Multiple-test correction

Multiple-test correction was applied using Sidak correction considering the number of effective tests. The number of effective tests was calculated following the formula described by ref. 60, which takes into account the correlation matrix of the measurements. This correction was applied to the commonly used genome-wide significance threshold, an alpha value of $5 \times 10^{-8}$. This resulted in a study-wide significance threshold, both genome-wide and study-wide alpha values and significance levels were used throughout.

### Functional annotation

Functional annotation of the QTL mapping results was performed using. FUMA was used with the default parameters to define independent SNPs ($r^2 < 0.6$), define independent loci (LD blocks closer than 500 kb), and annotate the functional consequences on genes. Phenoscanner[61,62] was used to find cQTL with reported eQTL effects. Genes linked to cQTL via an eQTL effect were inspected in the list of differentially expressed genes upon *Borrelia burgdorferi* s.l. stimulation.

### Comparison with pQTL from the UK Biobank

Available summary statistics from ref. 28 were downloaded via https://www.synapse.org/#!Synapse:syn51365301 on August 2023. The four proteins used in this study, IL-10, IL-1Ra, IL-6, and IL-1β, were examined. The p value for the top variants in all genome-wide significant cQTL was extracted and corrected using FDR.

### Differentially expressed genes upon *Borrelia Burgdorferi* stimulation

**PBMC isolation and stimulation.** Analyses on differentially expressed genes were performed in a cohort of healthy volunteers, who all provided written informed consent for venous blood sampling. Ethical permission for this study was approved by the Ethics Committee of the Radboud University Nijmegen (NL42561.091.12). Blood was collected in EDTA tubes (BD vacutainer). PBMCs were quickly isolated within 3 h

of collection. Blood was diluted with 1 volume of DPBS (Gibco, Thermo Fisher Scientific) before adding it to Ficoll-Paque (Pharmacia Biotech). Gradient centrifugation was performed for 30 min at $400 \times g$, using no brake. After centrifugation, the layer containing PBMCs was collected using a Pasteur pipette. PBMCs were washed twice with PBS, counted (BioRad cell counter), and adjusted to reach the final concentration of 2 million cells/ml in RPMI 1640 (Gibco, Thermo Fisher Scientific), supplemented with 10% heat-inactivated Fetal Cow Serum (Gibco, Thermo Fisher Scientific), gentamicin 10 mg/ml, L-glutamine 10 mM, and pyruvate 10 mM. Cells were seeded into wells to settle overnight before stimulation. To study PBMC transcriptomes upon live attenuated *B. burgdorferi* s.s. (ATCC strain 35210), PBMCs were stimulated at $1 \times 10^6$/mL. Cells were also incubated with RPMI 1640 only as a negative control. RNA was isolated from PBMCs at 4 and 24 h after stimulation.

**RNA isolation.** Cells were harvested and lysed in lysis buffer from the MirVanva MagMax RNA isolation kit (Applied Biosystems, Nieuwerkerk aan den IJssel, the Netherlands). RNA was isolated according to the manufacturer's instructions. RNA concentration was measured based on Optical density (OD260) using the Nanodrop machine (NanoDrop Technologies, Rockland, ME, USA). RNA integrity was determined using the Bioanalyzer (Agilent D2000). All samples had an RIN score >9.

**RNA sequencing and differentially expressed genes.** For PBMC sequencing, 1000 ng of total RNA (RIN score ≥9) were submitted for RNA library preparation using the NEXTflex TM Rapid Directional RNA-seq kit, BioScientific. NGS libraries were enriched for polyA tail RNA. Samples were sequenced using the Illumina NextSeq 500 platform, single-end read. Samples were randomly assigned into different flows and sequenced to reach 12–15 million reads per sample. Sequencing reads were then mapped to the human genome using STAR (version 2.3.0) with reference to Ensembl GRCh37.71. Read counts per gene was quantified by Htseq-count, Python package HTSeq (version 0.5.4p3) using the default union-counting mode (The HTSeq package, http://htseq.readthedocs.io/). Fragments were aligned using Hisat2. Raw counts were calculated with String Tie.

Differentially expressed genes were identified by statistical analysis using the DESeq2 package from Bioconductor. A statistically significant threshold (FDR $P \leq 0.05$ and fold change ≥2) was applied.

**Colocalization, Mendelian randomization, and fine mapping**
cQTLs that were found significant (FDR <0.05) in selected public summary statistics downloaded from the GWAS catalog[30] (last accessed on 18/08/2022) were tested for colocalization and Mendelian randomization (MR). Summary statistics were selected on two initially detected significant signals: age-related macular degeneration and allergic disease that were expanded to include a range of common infectious and non-infectious immune-mediated diseases (Suppl. Data 4). Colocalization was tested using the *coloc*[63] package in R, using the *coloc.abf* function, and a probability of H4 > 0.6 was considered indicative of association. MR was tested using the TwoSampleMR[39] package in R, in which the single-SNP MR was calculated using the default method, the Wald ratio test. A *p* value below 0.05 was considered a significant MR result. For multi-snp MR, SNPs with *p* value of associations below $10^{-5}$ were clumped at $R^2 < 0.001$ and used as instruments. Both inverse-variance weighted meta-analysis (multiplicative random effects) and MR Egger were used for the meta-analysis of the clumped SNPs. This was performed using the *mr* function from the TwoSampleMR package[39]. Since the MR Egger estimate for the intercept was not zero, this method was not used for causal estimation. IVW was the only method used for causal estimation.

Cochran's Q-test was applied to test for possible heterogeneity in the IVW estimates. MR Egger's intercept was used to inspect pleiotropy in the analysis. Both sensitivity measures are listed in Suppl. Data 9. Fine mapping was performed using the susieR package in R[64], using the *susie_rss* function, setting a maximum credible set size of 50 and using the correlation matrix of the SNPs in the loci.

**Differential effect of rs5743618 on IL-1Ra responses upon P3C and *Bb* stimulation**
To test for a different effect, we applied a linear regression model with the stimulant and the allelic dosage as an interaction term. We run the analysis for both time points, comparing P3C to *Bb* mix at a concentration of $10^5$ spirochetes/mL. We considered that a significant interaction term indicated a differential effect of the variant on cytokine production between the two stimulations.

**TLR1 validation cohort. Gene expression upon P3C stimulation**
A cohort of healthy individuals was recruited as part of the Cluster of Excellence RESIST (RESIST senior individuals (SI) cohort, https://www.resist-cluster.de/en/clinical-trials/resist-cohort/). A subset of the full cohort ($N = 100$), including individuals with a median age of 69 and 33% of female donors, was used to validate the genetic regulation of Pam3Cys responses. Peripheral blood mononuclear cells (PBMC) were isolated from human peripheral blood by density gradient centrifugation, and resuspended at a density of $2.5 \times 10^6$ cells/ml in Iscove's medium (Biochrom KG, Berlin, Germany) supplemented with 4% human heat-inactivated AB serum, 2 mM glutamine, 50 mg/ml of gentamicin, 100 mg/ml penicillin and streptomycin, and nonessential amino acids. $1 \times 10^6$ cells each were stimulated with 10 μg/ml Pam3Cys-Ser-(Lys)4 (EMC microcollections, Tübingen, Germany). Subsequently, RNA was extracted according to the manufacturer's protocols (Promega ReliaPrep). RNA concentration was determined by Qubit RNA High Sensitivity Assays (Thermo Fisher Scientific, Waltham, USA). Quality and integrity of total RNA was controlled on Agilent Technologies 2100 Bioanalyzer. mRNA was isolated using the Dynabeads mRNA DIRECT Kit (Thermo Fisher Scientific, Waltham, USA), and RNA sequencing libraries were generated from mRNA using NEBNext Ultra II Directional RNA Library Prep Kit (New England BioLabs) according to the manufacturer's protocol. Libraries were sequenced on a NovaSeq 6000 (Illumina, San Diego, USA) using the NovaSeq 6000 S2 Reagent Kit (paired-end sequencing of $2 \times 50$ bp) with an average of $5 \times 10^7$ reads per RNA sample. DNA was isolated from blood, and genotyping was performed at Erasmus UMC, Rotterdam, the Netherlands, using the GSA Array Infinium iSelect 24×1 HTS Custom Beadchip Kit (Illumina, San Diego, USA).

**Statistical analyses**
All statistical analyses were performed using R 4.0.1. The clustering of cytokine measurements was calculated using the Euclidean distance of Spearman's correlation values. PCA of cytokine measurements was calculated on an imputed matrix, via KNN imputation, where different normally distributed cytokine measurements are rows and donors are columns. Comparisons of cytokine values were performed using Spearman's correlation for quantitative measurements (age, C6 levels), the Wilcoxon rank-sum test for binary measurements (gender, disease phenotype), and the Wilcoxon signed-rank test for the treatment effect.

**Data visualizations**
All visualizations were obtained using R 4.0.1. and ggplot2 was used for plotting, unless indicated otherwise. Manhattan plot was made with the ggman package, heatmaps using ComplexHeatmap.

**Reporting summary**
Further information on research design is available in the Nature Portfolio Reporting Summary linked to this article.

## Data availability

The cytokine QTL summary statistics generated in this study have been deposited in the GWAS Catalog under study ids GCST90275731 to GCST90275790 and made available in a web tool at https://lab-li.ciim-hannover.de/apps/lyme_cqtl/. The Cytokine expression and C6 antibody ratios generated in this study are provided in Supplementary Data 1. The information on the start of antibiotic treatment is provided in Supplementary Data 2. The Genotype array data generated in this study are available in EGA under accession code EGAS50000000024.

## Code availability

All code used for the analyses is available at https://github.com/CiiM-Bioinformatics-group/Lyme_cQTL and at https://doi.org/10.5281/zenodo.10658534.

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

## Acknowledgements

The authors thank all volunteers of the LymeProspect cohort for their participation in the study. ERC Starting Grant (948207, Y.L.). Radboud University Medical Centre Hypatia Grant (2018) (Y.L.). European Union's Horizon 2020 research and innovation program under the Marie Skłodowska-Curie (grant agreement No 955321). Helmholtz Initiative and Networking Fund (1800167, C.-J.X.). Deutsche For-schungsgemeinschaft (DFG) Fund (497673685, C.-J.X.). ERC Advanced Grant (European Union's Horizon 2020 research and innovation pro-gram, grant agreement no. 833247, M.G.N.). Spinoza grant from the Netherlands Organization for Scientific Research (NOW) (M.G.N.). Netherlands Organization for Health Research and Development, (ZonMw) (project numbers 522001003, 522050003, 522050001, and 522050002). Dutch Ministry of Health, Welfare and Sport (VWS), Deutsche Forschungsgemeinschaft (DFG, German Research Founda-tion) through the Excellence Cluster RESIST (EXC 2155), under Ger-many's Excellence Strategy—EXC 2155—(project number 390874280). The charitable contributions raised by Rood voor Altijd and Minke Ver-strepen, donated through the AMC Foundation (Amsterdam UMC). INTERREG as part of the NorthTick project (J.W.H.).

## Author contributions

Y.L. and L.A.B.J. designed the study. J.B.-B. carried out the data analysis and interpretation. H.D.V. helped in the interpretation of the results. Y.L. and C.-J.X. supervised the analysis. J.B.-B and H.D.V drafted the initial manuscript. M.K.G., L.M.R., J.H., T.W., T.F.S., and Y.L. acquired the samples and RNAseq data of 100 donors upon P3C stimulation for validation. H.D.V., J.U., and C.C.v.d.W. recruited the patients and col-lected clinical samples and data as part of the LymeProspect project. L.A.B.J., J.W.H., B.-J.K., and C.C.v.d.W. conceived and designed the LymeProspect project. J.U., A.A., M.K.G., L.M.R., J.H., T.W., T.F.S., B.-J.K., C.C.v.d.W, H.t.H, C.-J.X., M.G.N., and J.W.H. revised and accepted the final manuscript.

## Funding

## Competing interests

The authors declare no competing interests.

## Additional information

**Supplementary information** The online version contains
supplementary material available at

Yang Li.

**Peer review information** *Nature Communications* thanks Kaur Alasoo,
Javeed Shah and the other, anonymous, reviewer(s) for their contribu-
tion to the peer review of this work. A peer review file is available.

