## [Peer Review File · Nature Communications]

A comprehensive genetic map of cytokine responses in Lyme borreliosisREVIEWER COMMENTS

Reviewer #1 (Remarks to the Author):

The authors Bete-Bataller et al present their evaluation of the host genetic determinants of cytokine responses to TLR and Borrelia in 1138 individuals that suffered from Lyme borreliosis (LB) at the time of diagnosis and longitudinally for a year after diagnosis. They measured 60 independent phenotypes from PBMC before and after antibiotic therapy and correlate findings with Lyme susceptibility as well as broader inflammatory disease susceptibility. Overall this study provides an interesting and potentially important addition to the literature, but is hampered by a lack of clarity regarding the validation of their observations and statistical corrections made for multiple comparisons, making interpretation of their conclusions challenging. In particular, it isn't clear to me what the appropriate multiple comparisons test would be given the GWAS of cQTL was performed >60 times with 2 time points. I believe that they used a higher threshold for statistical significance for these individuals, but it was not spelled out clearly in the text. A clear discussion of this issue is essential for proper consideration of their conclusions. The authors consider a genome-wide p value ($p < 5e-8$) as a significant value in this study, but they appear to use a nominal p value of 0.05 as their threshold for significance in their validation in the HFGP data. This validation should also be adjusted for multiple comparisons, as they are testing 5 - 25 SNPs in validation. Additionally, the authors use a validation cohort, but it is unclear where this data resides in the paper, and they do not appear to report all p values in validation.

Similarly, all tested p values for their validation for LB susceptibility from the referenced study should be provided to validate their data by demonstrating association between cQTL and gene expression data from Bb-stimulated PBMC. However, they do not show this data, nor does this fully validate any of their findings. It would enhance the quality of the paper for the authors to be clearer on when a specific SNP is being validated across studies, and when they are using these SNPs to evaluate immunologic phenotypes of interest via MR or others.

Major points:

1. Although the methods describe a validation set for the initial data, it is not well marked in the paper and I can't understand where it is shown. Please clearly mark the discovery and validation sets, and add a table with all p values for validation to permit appropriate evaluation of this dataset.
2. There are discrepancies in the sample size: the intro section states that there is a sample size of 1138, while figure 1a says 1060. Please clarify this result. There also needs to be mention of the clinical characteristics of these patients, including the proportion with severe/disseminated LB, their age, sex, and ethnicity.
3. Why weren't antibody and other non-cytokine measurements included in Figure 1b?
4. Line 149. Numbering for this subfigure is out of order and doesn't appear to describe what the finding is in the text.
5. Line 156. I do not understand the relationship between the cQTL and Bb-responding genes. Where were the Bb-responsive gene list taken from? Was this part of MSigDB's databases? Please provide more context and explanation for this observation.
6. Line 224. Validation of cQTL with the HFGP provides additional evidence of the activity of SNP blocks, but nominal p value 0.05 is inadequate. Please correct for multiple comparisons in the validation set of data.
7. Line 267. The authors state that the SCGB1D2 locus was associated with LB. What was the association of other LB loci with LB susceptibility? The complete association dataset should be added as a supplemental table. This dataset is currently incomplete as constituted.

Minor issues:

1. The font size for many subfigures was extremely small (Figure 1b-e, Figure 3, Figure 4b, c, e, Figure 5, Figure 6), making interpretation of these images difficult.

2. The text was confusing at places and would benefit from summary figures describing the gene effects seen, at least at a high level. Which cQTL were associated with susceptibility, which were associated with cytokine-induced cytokine responses?
3. For fine-mapping studies, is this SNP plausible for functionality given what is known about transcription factor biology? Does this SNP influence TF binding and/or predicted binding based on public datasets?
4. Line 277 -- this study is the largest cytokine QTL mapping study of what to date? Anyone? LB patients? This and multiple other sentences are confusing as the subject can be ambiguous
5. Line 283 -- please be specific with the locus that you are associating. SCGB1D2, yes?

Overall, this dataset is excellent and the findings are important and of broad value, but writing must be clearer throughout and paper organization must be improved to permit appropriate evaluation of the work, which currently precludes it from publication.

Reviewer #2 (Remarks to the Author):

This is a well written and mostly well-designed study, measuring cytokine profiles under multiple conditions (time points, stimulation, biological entity) in patient samples, highlighting potential new mechanisms of Lyme borreliosis (LB) progression. Besides connecting profiles with patient characteristics, they map the genetic structure of these cytokines and connect findings to previous studies using MR/coloc.

I have some comments I would like to be addressed.

General comment: Some of the results were difficult to evaluate due to the lack of information in table format or formatting of figures. In general the figure texts (labels/legends etc) needs to be made bigger as they are not readable at the moment. In particular Figure 5b-e (particularly second red dot in 5d) is unreadable.

1. I would like to see some motivation on why the particular cytokines were selected for profiling. Was this based on current literature on LB, if so this should be more clearly stated in e.g. the introduction.
2. As this study is based on patient samples, and the authors analyse some of the patient and general (age, sex etc) characteristics in relation to cytokine profiles I would like to see a Table included of patient characteristics. To interpret results at baseline it would also be helpful if proportions of patients having samples taken prior or post commencing Ab treatment (as the authors state that some differences are observed related to Ab treatment commenced or not).
3. To be able to interpret these results in light of other studies and in relation to the various stimulation I would be helpful to also see cQTLs in unstimulated conditions from the same patients. Particularly as the authors themselves note that stimulation is not the major driver of cytokine variation. Would the same cQTLs also have been detected in unstimulated state?
4. Tables of cQTLs in main text and Supplementary should include effect estimate and effect allele, as well as further information on e.g. allele frequency etc, so these results can be evaluated, utilised by others and interpreted in relation to previous studies.
5. Replication of findings using data from Li et al. The authors state that 7/9 associations could be replicated based on p-value, I assume that the direction of effect was also the same (however this information is not included in the manuscript). In figure S10 summarizing the replication, what does the row labels represent - it is difficult to interpret this figure without this information.

6. Are the presented results specific for this patient populations, stimulation, infection or can they also be replicated in studies with large enough samples of individuals? Besides replication in Li et al I would also like these results put into context of other large scale resources of protein QTLs (or cytokine GWASs as for e.g. IL6) in the general population (several such studies have been published in the last couple of years).

6. MR analyses cQTL - disease outcome:

- a) What criteria did the authors use for inclusion in the coloc/MR pipeline, a full list of outcomes/studies tested should be included in Supplementary Information. Did the authors also include GWAS of other infections?
- b) For significance in MR this should also be adjusted for multiple testing as per other analyses in the paper.
- c) I would like to see the result of the analyses (also including MR effect estimates, SNP used, outcome study) in table format e.g. in Supplementary.

Some minor comments:

Abstract line 56: English

Page 6, 188: missing word (significance?)

Fig S4: legend text labels colour as blue (looks green?)

Fig S7: poor quality image

Reviewer #3 (Remarks to the Author):

In this study, the authors measured the concentrations of four cytokines (IL-1b, IL-1Ra, IL-6 and IL-10) in 1060 Lyme borelliosis patients at two time points and in response to four stimuli. The authors further linked the cytokine measurements with genotype data collected from the same individuals. The bulk of the paper then focusses on identifying and interpreting genetic associations detected with cytokine levels. The association analysis presented in the paper mostly follows standard best practices and seems robust. I particularly like the fine mapping analysis performed at the TLR1-6-10 locus. However, I have some concerns about how the authors seek to link these associations to complex diseases, especially the Lyme disease. Finally, the utility of the results presented in the paper is strongly limited by almost complete lack of data sharing (i.e. genotype data not available and even association summary statistics "available upon request").

Major comments

1. On lines 137-138 you state that "cytokine-stimulation pairs were excluded for QTL mapping in case of non-normal distribution after log-transformation". While this is a valid strategy to avoid false positive associations driven by outlier measurements or other violations of linear model assumptions, it also excludes a significant proportion fo the data that could provide important insights. A standard procedure in GWAS studies to ensure that linear model assumptions are met is to use inverse normal transformation (INT) to force the data to have a normal distribution (see: <https://doi.org/10.1111/biom.13214>). While this can lead to some reduction in statistical power, it also enables the analysis of traits with skewed distributions with low risk of false positives. I would consider applying INT to your data before GWAS to also be able to analyse those cytokine-stimulation pairs that had non-normal distributions.
2. The authors have not justified why the summary statistics are only available upon request. Best practice in the field is to submit summary statistics to the GWAS Catalog. If this is not possible, then a clear justification for this should be given in the "Data availability" statement. Also, the authors should consider submitting their summary statistics to EGA if sharing them openly via GWAS Catalog is not possible.

3. On lines 214-216 you state that rs5743618-A does not alter IL-1Ra production upon P3C stimulation, but does so after Bb stimulation. However, the difference between the two stimuli seems rather small on Fig 4d. Can you quantify this with an interaction test? That is, is there a significant difference in the cQTL effect size between the P3C and Bb conditions? Or is your conclusion purely driven by the observation that you have significant association in one condition and not in the other (which could be driven by low power)?

4. In the section starting on line 220 ("Identified cQTL are relevant for other immune-mediated diseases" you present multiple analysis where you perform colocalisation analysis followed by single-variant mendelian randomisation. Could you please clarify what is the added value of single-variant MR in this instance? Significant colocalisation already implies that the same variant is strongly associated with both traits (otherwise PP0, PP1 or PP2 would be high) and thus their association betas are significantly different from zero. Unless there is no colocalisation, it is difficult to imagine a scenario where the ratio of these two betas ("the two-sample MR estimate") would not be significantly different from zero.

5. My primary concern lies with the analysis presented in section starting on line 260 ("Lyme borelliosis suceptibility") and on Fig 6. I feel that this section is highly speculative with very little solid genetic evidence and I would strongly consider excluding it from the paper. While the GWAS signal presented on Fig 6a is genome-wide significant, the cQTL signals on Fig 6b and antibody index signal on Fig 6c are extremely weak. Similarly to the analysis that you performed for IBD, can you please also test for colocalisation between the GWAS signal and the cytokine/antibody signals and report these results in the main text? If you do not see evidence for colocalisation then this should still be explicitly mentioned in the text. Furthermore, on Fig 6e-f and on Supplementary Fig 12 there is very large heterogeneity between the the different genetic instruments and the estimated "causal" effects from the IVW and MR Egger analyses are very different from each other. It seems that all of the IVW estimates are close to zero and most of the negative association is driven by one or two instruments with the others having null or opposite effects. Could you please discuss this in more detail in the paper? What other steps did you take to test that the genetic variants included in the analysis are valid instruments (i.e. satisfy MR assumptions)? Finally, for clarity the MR plots on Fig 6e-f and Supplementary Fig 12a should also include the origin (0,0 point). This would highlight even more clearly that the current MR Egger estimates seem implausible as they pass very far from the origin.

Minor comments

1. On line 172 you state that "... variants present in the loci had enough alternative alleles". This is confusing as it seems to indicate that you were focusing on multi-allelic variants that had multiple different alternative alleles. Based on the reading of the rest of the manuscript, it seems that you are actually referring to either minor allele count or minor allele frequency (MAF). Could you please clarify this in the text?

2. On lines 204-205, while discussing the TLR1-6-10 locus, you mention what Bb and P3C are both both TLR2 ligands. This is confusing as you do not explain in the text that TLR1 forms a complex with TLR2 and this is the reason why these stimuli are relevant. You do illustrate the complex on Fig 4f, but I would also recommend explicitly mentioning this assumption in the text.

3. The meaning of the dark and light blue lines on Fig 6e-f has not been explained. Similarly, on Fig 6e-f and Supplementary Figure 12 it is unclear what is exactly considered as outcome.

REVIEWER COMMENTS

Reviewer #1 (Remarks to the Author):

The authors Bete-Bataller et al present their evaluation of the host genetic determinants of cytokine responses to TLR and *Borrelia* in 1138 individuals that suffered from Lyme borreliosis (LB) at the time of diagnosis and longitudinally for a year after diagnosis. They measured 60 independent phenotypes from PBMC before and after antibiotic therapy and correlate findings with Lyme susceptibility as well as a broader inflammatory disease susceptibility. Overall this study provides an interesting and potentially important addition to the literature, but is hampered by a lack of clarity regarding the validation of their observations and statistical corrections made for multiple comparisons, making interpretation of their conclusions challenging. In particular, it isn't clear to me what the appropriate multiple comparisons test would be given the GWAS of cQTL was performed >60 times with 2 time points. I believe that they used a higher threshold for statistical significance for these individuals, but it was not spelled out clearly in the text. A clear discussion of this issue is essential for proper consideration of their conclusions. The authors consider a genome-wide p value ($p < 5e-8$) as a significant value in this study, but they appear to use a nominal p value of 0.05 as their threshold for significance in their validation in the HFGP data. This validation should also be adjusted for multiple comparisons, as they are testing 5 - 25 SNPs in validation. Additionally, the authors use a validation cohort, but it is unclear where this data resides in the paper, and they do not appear to report all p values in validation.

We thank the reviewer for the positive remarks. We would like to address the confusion about the different levels of significance used in the manuscript. We used two different significance levels, study-wide and genome-wide ones, as defined in Methods section 'Multiple test correction' (line 517).

Shortly, the higher significance level (study-wide) is calculated by adjusting for the number of effective tests, which is calculated based on the correlation of the phenotypes measured. Study-wide significant cQTL are described in Figure 2b. This threshold was applied to control for multiple testing while adjusting for the correlation in the phenotypes¹. Recent studies report this as a threshold to define statistical significance in genome-wide association studies^{2,3}.

The genome-wide significance threshold was also explored as those SNPs might be of potential biological relevance. These cQTL are proven to be immunologically relevant, such is the case of the *cis-IL10* and *CFH* loci, and as is proven by the enrichment of *Borrelia Burgdorferi* (*Bb*) responding genes in genome-wide significant cQTL associated upon *Bb* stimulation. Therefore, we considered both sets of cQTL in our study of immune-mediated diseases and Lyme Borreliosis. Recent publications do also report both significance thresholds, interpreting the signals found in both³.

To avoid any potential confusion about these two levels of significance, we have adjusted the text in the corresponding methods section to 'This resulted in a study-wide significance threshold, both genome-wide and study-wide alpha values and significance levels were used

throughout' (line 522). We have addressed the multiple-test correction in the replication in a healthy of cQTL in the corresponding point of the Major points 6.

Similarly, all tested p values for their validation for LB susceptibility from the referenced study should be provided validate their data by demonstrating association between cQTL and gene expression data from *Bb*-stimulated PBMC. However, they do not show this data, nor does this fully validate any of their findings. It would enhance the quality of the paper for the authors to be clearer on when a specific SNP is being validated across studies, and when they are using this SNPs to evaluate immunologic phenotypes of interest via MR or others.

We thank for the reviewer for raising this important question about two different datasets used to complement the cQTL found in the current study.

The recent study on LB susceptibility⁴ (Strausz et al.), performed in the Finnish population revealed novel LB susceptibility loci. We hypothesized that our identified cQTLs could provide a functional interpretation for LB GWAS signals, similarly to the use of other molecular QTLs, such as eQTLs⁵. Indeed, we found that the SCGB1D2 locus, one of the two loci identified in the Finnish cohort, regulated cytokine production and antibody levels in the LymeProspect cohort. We also wanted to clarify that here we did not intend to replicate lyme GWAS result or replicat our cQTLs in the lyme GWAS data.

The gene expression data from *Bb*-stimulated PBMCs was generated to prioritize the causal genes within the identified cQTL loci. As there are usually multiple genes within the locus of each cQTL, it is challenging to pinpoint the most relevant genes. Therefore, we measured gene expression in unstimulated and *Bb*-stimulated PBMCs and identified *Bb*-responding genes by performing differential expression between stimulated and unstimulated samples.

Moreover, we aimed to use *Bb*-responding genes to validate the relevance of the cQTL genes in the response to *Bb* in general, where the significant enrichment indicates that cQTL are pointing at relevant genes and putatively functional effects in the response to *Bb*. To make clearer which specific cQTL contain *Bb* responding genes, we have modified the column that indicated this in Figure 2b and Extended Data Table 2 from 'DE' to '*Bb* responding'. The LB susceptibility locus is not a study or genome-wide significant cQTL and therefore not part of either of the tables. To access this information readers may extract the GWAS data from the cited Strausz et al., 2022 and the cQTL data from the GWASCatalog, containing the full summary statistics for the current study.

Major points:

1. Although the methods describe a validation set for the initial data, it is not well marked in the paper and I can't understand where it is shown. Please clearly mark the discovery and validation sets, and add a table with all p values for validation to permit appropriate evaluation of this dataset.

We thank the reviewer for raising this point and we apologize for the confusion.

In the original manuscript, the term 'Validation cohort' identified a cohort of 100 healthy individuals in which both genetics and RNA-seq data upon Pam3Cys and *Bb* stimulation were measured. Here, we aimed to validate the cQTL locus (TLR-1) which was associated with the production of all four cytokines (IL-10, IL-1Ra, IL-6 and IL-1 β) upon *Bb*-mix stimulation and all cytokines except IL-1Ra upon P3C stimulation. The gene expression results in the 'Validation cohort' showed changes consistent to the observed in the LB cohort, reflecting a differential regulation of IL-1Ra responses upon P3C.

To avoid potential confusion, we have edited the text to name it 'TLR1-locus Validation cohort'. Following the reviewer's recommendation, we have now also listed the p-values calculated in the 'TLR1-locus Validation cohort' in Extended Data Table 4.

2. There are discrepancies in the sample size: the intro section states that there is a sample size of 1138, while figure 1a says 1060. Please clarify this result. There also needs to be mention of the clinical characteristics of these patients, including the proportion with severe/disseminated LB, their age, sex, and ethnicity.

We thank the reviewer for spotting the differences in cohort size. These discrepancies are the result of a different cohort definition.

Initially, the cohort of the LymeProspect project (see also Ursinus et al., 2021 The Lancet Regional Health) was described, in which 1,138 participants with Lyme Borreliosis were recruited. For the cQTL analysis, cytokine production upon stimulation and genotype was measured for 1,060 samples.

We have now clarified this on line 90, indicating that 'Whole-genome assessment of genetic variation was performed, and antibody concentrations and cytokine responses to various stimuli were measured in 1,060 participants.

In addition, following the reviewer's request, the clinical characteristics of the patients have been included in an Extended Data Table (Extended Data Table 1).

3. Why weren't antibody and other non-cytokine measurements included in Figure 1b?

We thank the reviewer for the suggestion.

The initial idea of assessing the inter-correlation the cytokine profiles was to examine whether the different stimulations or cytokines would cluster separately. Therefore, in Figure 1b, we focus on evaluating the inter-regulation of cytokine responses and the main driver of variation in the system.

We had assessed the correlation of cytokine response profiles with antibody and other non-cytokine measurements (as the reviewer suggests above) in Figure 1e. There, we found a significant negative correlation between IL-10 production and antibody levels.

4. Line 149. Numbering for this subfigure is out of order and doesn't appear to describe what the finding is in the text.

We thank the reviewer for raising this point.

We understand that the subfigure numbering can be misinterpreted. In the text we are describing one of the study-wide significant loci, EFR3A locus. Information on how the locus is associated to different cytokine responses is shown in locus 19 in Figure 3a, and its regulation of IL-1 β is shown in Extended Data Fig. 5d-f. To improve the understanding of this numbering, we have added the names of the relevant loci (including EFR3A), cited in the main text, to the bottom of the heatmap in Figure 3a.

5. Line 156. I do not understand the relationship between the cQTL and Bb-responding genes. Where were the Bb-responsive gene list taken from? Was this part of MSigDB's databases? Please provide more context and explanation for this observation.

We thank the reviewer for raising this point.

CQTLs prioritize genomic regions exhibiting significant correlations between genetic variants and cytokine responses. Within each of these genomic regions, numerous genes are present, making it challenging to pinpoint the candidate genes. To address this, performing a differential gene expression analysis by comparing samples under baseline and stimulation conditions within the cQTL locus can identify genes with significantly different expression levels between these groups. These differentially expressed genes are more likely to be the causal genes within the cQTL locus.

In the original analysis, we used PBMCs from healthy volunteers, which were stimulated with *Bb* for 4 or 24 hours and compared to unstimulated PBMCs. Bb-responding genes were defined as differentially expressed genes after *Borrelia burgdorferi* stimulation. These methods were described in the Methods section 'Differentially expressed genes upon *Borrelia burgdorferi* stimulation'. To make sure that this gene list is clearly understood, we have modified line 170 to 'We therefore identified as Bb-responding genes those differentially expressed compared control-stimulated PBMCs,'.

6. Line 224. Validation of cQTL with the HFGP provides additional evidence of the activity of SNP blocks, but nominal p value 0.05 is inadequate. Please correct for multiple comparisons in the validation set of data.

We thank the reviewer for the suggestion and fully agree with the reviewer. To address this, we are now testing the validation of the identified genome-wide cQTL by applying multiple-test correction (FDR) as well as reporting their raw p values on Extended Data Figure 10.

Based on the new results, we have now modified the text to reflect the different levels of validation. The text on line 275 now reads: "We found 2/3 study-wide significant (loci 7, 16; Extended Data Fig. 10a) and 1/6 genome-wide significant (locus 3; Extended Data Fig. 10b) variants were significantly associated with cytokine responses in the HFGP (FDR<0.05). At

the nominal threshold of 0.05, 5/6 genome-wide significant variants were significant (Extended Data Fig. 10b).

Notably, the top variant in locus 3, which regulates IL-10 responses in LB, showed an opposite effect in IL-6 and IL-1 β in the HFGP (Extended Data Fig. 10b), coinciding to what we observed in LB (Figure 3a, locus 3). IL10 is anti-inflammatory, IL6 and IL1b are pro-inflammatory. All in all, we found a moderate replication in healthy of the cQTL found in LB”.

7. Line 267. The authors state that the SCGB1D2 locus was associated with LB. What was the association of other LB loci with LB susceptibility? The complete association dataset should be added as a supplemental table. This dataset is currently incomplete as constituted.

We fully agree with the idea of comparing the cQTL with LB susceptibility loci. This was one of the central ideas in the original manuscript, where we were able to link the effect of variation in the SCGB1D2 locus to immune function.

The association between LB susceptibility and the SCGB1D2 locus is a result of the GWAS performed as part of the Finngen study. This is currently described by their authors in a preprint (Strausz et al. 2022 bioRxiv). We were therefore able to provide the association between variation in one of the LB susceptibility loci and cytokine responses. The full dataset of LB susceptibility loci is therefore not part of our study but belonging to the authors in Strausz et al.

Minor issues:

1. The font size for many subfigures was extremely small (Figure 1b-e, Figure 3, Figure 4b, c, e, Figure 5, Figure 6), making interpretation of these images difficult.

We thank the reviewer for their comment. We have now increased the font size in the figures requested.

2. The text was confusing at places and would benefit from summary figures describing the gene effects seen, at least at a high level. Which cQTL were associated with susceptibility, which were associated with cytokine-induced cytokine responses?

We understand the question raised by the reviewer. It seems that the term cQTL might be confusing. cQTL, cytokine quantitative trait loci, are those genomic loci that we found associated with stimulation-induced cytokine responses. This is done by measuring cytokine production after stimulation and mapping cytokine levels to common genetic variants using linear modelling. Per se, cQTL are not associated with susceptibility, only with stimulation-induced cytokine responses. This is how cQTL are primarily found and then, thereafter, these loci can also be found associated with other traits like infection susceptibility. Through the manuscript, we make use of those associations to test for causal relationships.

3. For fine-mapping studies, is this SNP plausible for functionality given what is known about

transcription factor biology? Does this SNP influence TF binding and/or predicted binding based on public datasets?

We understand that the reviewer is referring to the top result of our fine-mapping of the TLR1-6-10 locus. In this case, rs5743618 is the top selected SNP. Since rs5743618 is in the coding region of *TLR1* and has a missense effect on the protein sequence, we can imply that it exerts its regulatory effect through affecting protein function. Therefore, transcription factor biology would not add new insights into the potential regulatory effects of rs5743618. If instead we are considering fine-mapping studies in general, adding public data on TF binding could be beneficial, similar to public data on the variant effect on protein sequence.

4. Line 277 -- this study is the largest cytokine QTL mapping study of what to date? Anyone? LB patients? This and multiple other sentences are confusing as the subject can be ambiguous

We agree with the reviewer, this point might be confusing. It is the largest cytokine QTL mapping study of any phenotype (Healthy, LB patients or other disease phenotype) to date. We have fixed this in the text now by writing: "the largest cytokine QTL mapping performed on any population to date" on line 350.

5. Line 283 -- please be specific with the locus that you are associating. SCGB1D2, yes? We have added the locus name and the citation to the previous study. It now reads "was previously described to affect LB susceptibility, SCGB1D2, to cQTL."

Overall, this dataset is excellent and the findings are important and of broad value, but writing must be clearer throughout and paper organization must be improved to permit appropriate evaluation of the work, which currently precludes it from publication.

We thank the reviewer for their remarks. All points have been taken care of, and we hope the new manuscript version allows for a more intuitive understanding of the relevance of the work.

Reviewer #2 (Remarks to the Author):

This is a well written and mostly well-designed study, measuring cytokine profiles under multiple conditions (time points, stimulation, biological entity) in patient samples, highlighting potential new mechanisms of Lyme borreliosis (LB) progression. Besides connecting profiles with patient characteristics, they map the genetic structure of these cytokines and connect findings to previous studies using MR/coloc.

I have some comments I would like to be addressed.

General comment: Some of the results were difficult to evaluate due to the lack of information in table format or formatting of figures. In general the figure texts

(labels/legends etc) needs to be made bigger as they are not readable at the moment. In particular Figure 5b-e (particularly second red dot in 5d) is unreadable.

We thank the reviewer for this. We have now modified all manuscript figures to be sure that all figure texts are readable.

1. I would like to see some motivation on why the particular cytokines were selected for profiling. Was this based on current literature on LB, if so this should be more clearly stated in e.g. the introduction.

We agree and thank the reviewer for raising this point.

IL-6, IL-10, IL-1Ra and IL-1 β are the main cytokines produced during the innate response to *Bb*. All four therefore contribute to the subsequent adaptive response and their regulation is determinant for the development of the disease. Moreover, they have been linked to differences in the clinical presentation of the disease that we could also observe in Fig1d.

We have added an explanation to the introduction on the selection of the cytokines and stimulations. Line 93 now reads "To profile cytokine production, four interleukins, IL-6, IL-10, IL-1Ra and IL-1 β , were measured upon 24-hour stimulation with inactivated pathogens. IL-6 and IL-1 β are known to mediate the innate response to *Bb*²⁰. IL-1Ra regulates IL-1 signaling and is known to be less induced by *Bb* than its pro-inflammatory part, IL-1 β ^{21,22}. IL-10 exerts anti-inflammatory effects that regulate the adaptive response to *Bb*^{5,14}".

2. As this study is based on patient samples, and the authors analyse some of the patient and general (age, sex etc) characteristics in relation to cytokine profiles I would like to see a Table included of patient characteristics. To interpret results at baseline it would also be helpful if proportions of patients having samples taken prior or post commencing Ab treatment (as the authors state that some differences are observed related to Ab treatment commenced or not).

This is a very important point that we had missed. We have added a table now: Extended Data Table 1. There, we summarize the basic information that was used through the manuscript and the number of samples taken prior and post start of the Antibiotic treatment.

	LB patients (n=1,059)
Male sex - no (%)	435 (41.1%)
Age (years)	
median [min, max]	55 [19, 87]
mean (SD)	53.5 (12.5)
LB manifestation	
EM	1,017 (96%)
Disseminated LB	42 (4%)
Ancestry	European

Blood taken before the start of antibiotic treatment?

Yes

No

128 (13.1%)

847 (86.9 %)

Extended Data Table 1. Main characteristics of the LymeProspect cohort individuals included in the analyses.

3. To be able to interpret these results in light of other studies and in relation to the various stimulation I would be helpful to also see cQTLs in unstimulated conditions from the same patients. Particularly as the authors themselves note that stimulation is not the major driver of cytokine variation. Would the same cQTLs also have been detected in unstimulated state?

We agree with the reviewer and think that this is an interesting point. We have focused on cQTLs upon stimulation, as it replicates the in vivo immune response. Cytokine concentrations that are found before in unstimulated settings, such as the RPMI stimulation in our data, are mainly derived from circulating cytokines and therefore reflect other systemic inflammatory processes. Moreover, most of the cytokines measured are found at negligible levels in circulation, which indicates that values observed in unstimulated settings could be a result of contamination. Because of all of this, in this study we focused on understanding the inter-individual variation in the regulation of immune function by mapping cytokine production upon stimulation. We did, however, attempt to map the cytokine levels upon control stimulation (RPMI) and found insufficient measures to map to the genome except for the case of IL-1Ra. We have not used these results in the current manuscript as we think they are out of the scope of the current study, in which we aim to decipher the regulation of responses to pathogens.

4. Tables of cQTLs in main text and Supplementary should include effect estimate and effect allele, as well as further information on e.g. allele frequency etc, so these results can be evaluated, utilised by others and interpreted in relation to previous studies.

We thank the reviewer for the remark. We have adjusted this now by adding the effect estimate, the effect allele, and the effect allele frequency. This way, we hope our data can be used by others. All the relevant data will now also be available via the GWASCatalog and a webserver. Changes can be seen in Figure 2b and Extended Data Table 2.

b

Locus Nr.	Lead SNP							Cytokine	Stimulation	Time	Cell System	eQTL	Bb-responding
	SNP	Chromosome	Position	REF	ALT	ALT_AF	P						
7	rs6815814	4	38816338	A	C	0.24198	3.35e-28	IL-10, IL-6, IL-1b, IL-1ra	Bbmix 1e5, Bbmix MOI 3, Bbmix MOI 10, P3C, Bbmix MOI 30	After treatment, Baseline	PBMC, whole blood	RP11-617D20.1, AC021860.1, TLR1, FLJ13197, TLR10, TLR6, KLF3, KLHL5, TMEM156, FAM114A1, NCS1, TAS2R31, TAS2R45, AKR1C2, LOC100653286, CERS4, RPL9	TLR1, TLR6, FAM114A1, RP11-617D20.1, AC021860.1, NCS1
10	rs145818098	5	1197036	A	G	0.98969	4.37e-10	IL-10, IL-6	LPS, Bbmix MOI 30	After treatment	whole blood		
16	rs35345753	7	22740513	C	G	0.21178	2.65e-11	IL-6	C.albicans, Bbmix MOI 30, Bbmix MOI 10, LPS	After treatment, Baseline	PBMC, whole blood	TOMM7, IL6, AC005682.5, KLHL7-AS1, NUPL2, TARDBP, MASP2, LOC643387, AC006026.13, STEAP1B	IL6, STEAP1B
19	rs6990239	8	132831672	C	T	0.08891	1.49e-11	IL-1b	Bbmix 1e5	Baseline	PBMC	EFR3A	
27	rs76009888	13	51945061	C	T	0.97685	1.24e-09	IL-6	LPS	After treatment	whole blood	INTS6-AS1, RPS4XP16, INTS6	

Figure 2. Genome-wide and study-wide significant cytokine QTL. **b**, Table of study-wide significant loci ($p < 1.5 \times 10^{-9}$). Each locus is described by its top SNP and cytokine-stimulation for which it is found. The eQTL column indicates expression QTL effects, extracted from the eQTLgen consortium. Genes are labelled as 'Bb-responding' if they are also responding to *B. burgdorferi*.

5. Replication of findings using data from Li et al. The authors state that 7/9 associations could be replicated based on p-value, I assume that the direction of effect was also the same (however this information is not included in the manuscript). In figure S10 summarizing the replication, what does the row labels represent - it is difficult to interpret this figure without this information.

We thank the reviewer for raising this point. We agree and have now changed Extended Data Fig. 10. We have now adjusted for multiple tests, as also requested by #Reviewer1 and have also colored the heatmap indicating whether the direction of effect matches between the current cQTL on LB and the cQTL on Health from Li et al. Multiple test correction now reveals a moderate replication: the effect direction matches in all significant cases after correction except for the one in Locus 3. This case is, however, still matching our interpretations. Locus 3 regulates IL-10 responses (anti-inflammatory) in our data and the negative direction in Li et al. is found for pro-inflammatory responses (IL-8, IL-6 and IL-1 β). We can observe the same discordant regulation in our data (Fig 3a, locus 3), where we observed positive effects in IL-10 responses and negative effects in IL-6 responses. We have

also now added 'Locus' before the loci numbers to indicate that the row labels represent the loci numbers.

The text on line 275 now reads: "We found 2/3 study-wide significant (loci 7, 16; Extended Data Fig. 10a) and 1/6 genome-wide significant (locus 3; Extended Data Fig. 10b) variants were significantly associated with cytokine responses in the HFGP (FDR<0.05). At the nominal threshold of 0.05, 5/6 genome-wide significant variants were significant (Extended Data Fig. 10b).

Notably, the top variant in locus 3, which regulates IL-10 responses in LB, showed an opposite effect in IL-6 and IL-1 β in the HFGP (Extended Data Fig. 10b), coinciding to what we observed in LB (Figure 3a, locus 3). IL10 is anti-inflammatory, IL6 and IL1 β are pro-inflammatory. All in all, we found a moderate replication in healthy of the cQTL found in LB".

Extended Data Figure 9. Replication of cQTL in a healthy cohort. Summary statistics extracted from Li et al., Cell, 2016. See reference for methods. * P<0.05 ** FDR<0.05 *** P<5x10⁻⁸. a, Study-wide significant loci, b, Genome-wide significant loci.

6. Are the presented results specific for this patient populations, stimulation, infection or can they also be replicated in studies with large enough samples of individuals? Besides replication in Li et al I would also like these results put into context of other large scale resources of protein QTLs (or cytokine GWASs as for e.g. IL6) in the general population (several such studies have been published in the last couple of years).

We thank the reviewer for the useful suggestion. As the reviewer points out, replication in the HFGP (Li et al.) can confirm that the top, most significant signals are replicated in a smaller cohort of healthy individuals. The reviewer suggests comparing our results with existing protein QTLs. These studies are mainly centered in circulating plasma protein concentrations, which are biomarkers for underlying process that differ from the functional data that was used in this study. However, following the reviewer's suggestion, we thought it would be interesting to explore the overlap between the cQTL found in this study and a recent protein QTL study, the recently carried out by the UK biobank. We have summarized this in Extended Data Fig. 7. We followed the same approach we took to replicate our cQTLs in the HFGP. We found one significant (FDR<0.05) pQTL at locus 8, which is associated with IL-10 responses to LPS in our data and circulating IL-10 levels in the UKBB. Interestingly, we found a significant association (P<0.05) before multiple test correction between locus 3, the IL-10 locus, and IL-10 plasma levels (P<0.05) in the UKBB. This could be a cis regulation that affects both plasma protein levels and IL-10 production upon a challenge. However, the signal is weak, which could also make the case of a distinct regulation of homeostatic levels and response to infection. We have now added this to the results section on line 209,

“Comparing our results with a recent QTL study on circulating proteins in the UK Biobank²⁵, we found a significant (FDR<0.05) association between Locus 8 (associated with IL-10 production upon LPS stimulation) and circulating IL-10 levels (Extended Data Fig. 14). This could reflect an SNP effect both in circulating and in functional production of IL-10”. Methods used have now been added to the methods section “Comparison with pQTL from the UK Biobank”.

6. MR analyses cQTL - disease outcome:

a) What criteria did the authors use for inclusion in the coloc/MR pipeline, a full list of outcomes/studies tested should be included in Supplementary Information. Did the authors also include GWAS of other infections?

We thank the reviewer for pointing this out. We have now added a table, Extended Data Table 5, in which readers will be able to find the accession codes and the trait as it is described on the GWASCatalog. Two diseases were initially selected to run *coloc* and MR based on an initial search in Phenoscanner: Age related macular degeneration and allergic disease. This list was expanded by including common immune-mediated diseases and infections for which the full summary statistics was available on the GWASCatalog. We have specified this in the methods. GWAS for other infections was inspected for the traits ‘Bacterial infection’, COVID, CMV seropositivity, Hepatitis C, Herpes seropositivity, HIV infection, Measles, Mumps and Tuberculosis. However, only ‘Bacterial infection’, COVID, Hepatitis C and HIV infection had available results for any of the 34 variants we tested. None of them had a significant association with the variants tested after FDR correction. Such information has been added to line 583: “Summary statistics were selected on two initially detected significant signals: age related macular degeneration and allergic disease that were expanded to include a range of common infectious and non-infectious immune-mediated diseases (Extended Data Table 4).”

b) For significance in MR this should also be adjusted for multiple testing as per other analyses in the paper.

We thank the reviewer for the comment. We agree and have now changed the original ‘suggestive significant’ threshold of $1e-4$ to an alpha of 0.05 after FDR correction. Only the four pairs that were significant are then tested for coloc and MR. We have adjusted the text and Fig 5a accordingly on line 285: “ In total, we observed significant (FDR<0.05) associations in four loci (Fig. 5a).”

c) I would like to see the result of the analyses (also including MR effect estimates, SNP used, outcome study) in table format e.g. in Supplementary.

We thank the reviewer for the suggestion. Extended Data Table 6 now details effect estimates as well as exposure, outcome and p-values.

Some minor comments:

Abstract line 56: English

We thank the reviewer for the suggestion, but we were not able to find the mistake they are referring to. Could they specify what part of the writing is incorrect?

Page 6, 188: missing word (significance?)

We have now added 'significance'.

Fig S4: legend text labels colour as blue (looks green?)

We thank the reviewer for the comment. We have now added the plot legend, increased the quality of the image, and modified the figure legend to make this distinction as clear as possible.

Fig S7: poor quality image

We thank the reviewer for the comment. We have now updated the image to increase its quality.

Reviewer #3 (Remarks to the Author):

In this study, the authors measured the concentrations of four cytokines (IL-1b, IL-1Ra, IL-6 and IL-10) in 1060 Lyme borelliosis patients at two time points and in response to four stimuli. The authors further linked the cytokine measurements with genotype data collected from the same individuals. The bulk of the paper then focusses on identifying and interpreting genetic associations detected with cytokine levels. The association analysis presented in the paper mostly follows standard best practices and seems robust. I particularly like the fine mapping analysis performed at the TLR1-6-10 locus. However, I have some concerns about how the authors seek to link these associations to complex diseases, especially the Lyme disease. Finally, the utility of the results presented in the paper is strongly limited by almost complete lack of data sharing (i.e. genotype data not available and even association summary statistics "available upon request").

Major comments

1. On lines 137-138 you state that "cytokine-stimulation pairs were excluded for QTL mapping in case of non-normal distribution after log-transformation". While this is a valid strategy to avoid false positive associations driven by outlier measurements or other violations of linear model assumptions, it also excludes a significant proportion fo the data that could provide important insights. A standard procedure in GWAS studies to ensure that linear model assumptions are met is to use inverse normal transformation (INT) to force the data to have a normal distribution (see: <https://doi.org/10.1111/biom.13214>). While this can lead to some reduction in statistical power, it also enables the analysis of traits with skewed distributions with low risk of false positives. I would consider applying INT to your data before GWAS to also be able to analyse those cytokine-stimulation pairs that had non-normal distributions.

We thank the reviewer for the suggestion. During the pre-processing of the cytokine measures we considered the use of inverse normal transformation (INT) prior to QTL mapping. We decided that log transformation enabled us to keep a majority of the measures and maintain the statistical power. Another factor we considered is the nature of the ELISA measurements. ELISA values used in this study were extrapolated using a standard curve, with upper and lower limits of detection depending on the antibody. Values outside the range of the detection were imputed to the corresponding limit leading to ties in the data that are not salvageable by applying INT. Non-normally distributed variables contained many of the mentioned ties (As can be seen in Extended Data Fig. 4). For this analysis, we decided to run the mapping exclusively in those variables exhibiting a normal distribution.

2. The authors have not justified why the summary statistics are only available upon request. Best practice in the field is to submit summary statistics to the GWAS Catalog. If this is not possible, then a clear justification for this should be given in the "Data availability" statement. Also, the authors should consider submitting their summary statistics to EGA if sharing them openly via GWAS Catalog is not possible.

We completely agree with this reviewer's comment. We had initially added this statement while working out the best option to make the summary statistics available. At this moment, the summary statistics for all cytokine stimulations have been uploaded to the GWAS Catalog (GCP000694) and will be publicly available upon publication. Moreover, we have created a shiny app server that will allow any reader to quickly inspect the significant associations. The app will be up and running as soon as possible.

3. On lines 214-216 you state that rs5743618-A does not alter IL-1Ra production upon P3C stimulation, but does so after Bb stimulation. However, the difference between the two stimuli seems rather small on Fig 4d. Can you quantify this with an interaction test? That is, is there a significant difference in the cQTL effect size between the P3C and Bb conditions? Or is your conclusion purely driven by the observation that you have significant association in one condition and not in the other (which could be driven by low power)?

We thank the reviewer for the suggestion. Initially, we hypothesized there could be a different regulation due to the absence of a significant association in IL-1Ra upon P3C stimulation in both timepoints, together with a significant negative effect that can be observed in Fig 4d bottom left and the replication in an independent cohort. We have now also tested this using an interaction term between stimulation and allelic dosage for rs5743618 as suggested. The methods are detailed at section "Differential effect of rs5743618 on IL-1Ra responses upon P3C and Bb stimulation". The results can be found in Extended Data Table 3. We found that, at both timepoints, the effect estimate of rs5743618 on IL-1Ra levels is significantly different between P3C and Bb. By incorporating these results, we can have greater confidence in the distinct regulation of IL-1Ra responses to P3C. Among the four cytokines studied, this is the only one showing negative regulation. This finding underscores the need for further investigation into this mechanism. Especially, considering the significance of rs5743618 in pathogen recognition, its prevalence among individuals of European ancestry, and the evidence of positive selection.

We have now modified the text on line 239 “We observed that the positive effect of rs5743618-A on IL-1Ra production upon *Bb* stimulation was significantly different upon P3C stimulation (Fig. 4c,d, Extended Data Fig. 9a-d, Extended Data table 3, $P_{\text{interaction-Baseline}}=5.77\times 10^{-7}$, $P_{\text{interaction-After treatment}}=2.5\times 10^{-6}$), indicating a differential genetic differential genetic regulation between *Bb* and P3C stimulation”.

4. In the section starting on line 220 (“Identified cQTL are relevant for other immune-mediated diseases” you present multiple analysis where you perform colocalisation analysis followed by single-variant mendelian randomisation. Could you please clarify what is the added value of single-variant MR in this instance? Significant colocalisation already implies that the same variant is strongly associated with both traits (otherwise PPO, PP1 or PP2 would be high) and thus their association betas are significantly different from zero. Unless there is no colocalisation, it is difficult to imagine a scenario where the ratio of these two betas (“the two-sample MR estimate”) would not be significantly different from zero.

We thank the reviewer for this comment. We understand the concerns and agree with the fact that colocalization and single-SNP MR yield very similar results in this case. Our rationale was to first apply colocalization to confirm that the same locus and variants are significantly associated with both traits, to then apply MR as an alternative method that also tests for directionality. Since both methods are widely used in the field, we thought it was pertinent to report the results for both. This way, we can report the directionality in Fig. 5f while providing evidence for its significance. To better indicate that colocalization and single-SNP MR have the same added value, we have now changed the results section parts in which we claimed that MR “confirmed” the coloc results to reporting both MR and coloc results. This can be read starting on line 290.

5. My primary concern lies with the analysis presented in section starting on line 260 (“Lyme borelliosis suceptibility”) and on Fig 6. I feel that this section is highly speculative with very little solid genetic evidence and I would strongly consider excluding it from the paper. While the GWAS signal presented on Fig 6a is genome-wide significant, the cQTL signals on Fig 6b and antibody index signal on Fig 6c are extremely weak. Similarly to the analysis that you performed for IBD, can you please also test for colocalisation between the GWAS signal and the cytokine/antibody signals and report these results in the main text? If you do not see evidence for colocalisation then this should still be explicitly mentioned in the text.

We thank the reviewer for the comments and their concerns on the results presented on Fig 6. Fig 6 is detailing the first ever use case for cQTL. Here, we aimed to use the functional genomics data that is available from the LymeProspect to better understand the mechanisms that associate the SCG1BD2 locus with Lyme Borrelia susceptibility. We found extremely relevant for those studying Lyme Borrelia and due to the limited knowledge on the determinants of susceptibility to report that the risk allele was associated with higher inflammation (specifically, higher IL-10 and lower IL-6) and higher antibody levels. To secure that these associations are not coincidental, we are now reporting a stronger signal that we found in cQTL: IL-6 to IL-10 ratio, as opposed to examining each cytokine separately, along with antibody levels. Following the reviewer’s suggestion, we have now also performed colocalization for all three measures: Inflammatory ratio at both

timepoints and antibody levels. Fig 6a now shows the strong evidence for colocalization we found for both ratio ($PP.H4_{\text{Baseline, After treatment}} = 0.97, 0.7$) and Antibody levels ($PP.H4_{\text{After treatment}} = 0.74$). In all three cases, the missense variant rs2232950 was the one with the higher evidence for colocalization (Extended Data Table 8). We have now added this as evidence for the association between the locus, and specifically rs2232950, and inflammation. Fig 6 has been adjusted and the text at the results (line 333 onwards) modified to add the extra evidence colocalization brings. Colocalization results at locus level and SNP level are found in Extended Data Table 8 and 9.

Furthermore, on Fig 6e-f and on Supplementary Fig 12 there is very large heterogeneity between the the different genetic instruments and the estimated "causal" effects from the IVW and MR Egger analyses are very different from each other. It seems that all of the IVW estimates are close to zero and most of the negative association is driven by one or two instruments with the others having null or opposite effects. Could you please discuss this in more detail in the paper? What other steps did you take to test that the genetic variants included in the analysis are valid instruments (i.e. satisfy MR assumptions)? Finally, for clarity the MR plots on Fig 6e-f and Supplementary Fig 12a should also include the origin (0,0 point). This would highlight even more clearly that the current MR Egger estimates seem implausible as they pass very far from the origin.

We thank the reviewer for the helpful comment. We have now run heterogeneity and horizontal pleiotropy tests to account for this in the analysis (Extended Data Table 9).

To satisfy MR assumptions, we selected independent variants with a p-value below $1e-5$ and MAF above 0.05, we tested for heterogeneity and horizontal pleiotropy. Now all plots include the origin of coordinates to show that the current MR Egger estimates are passing through the origin, and we can therefore consider them valid.

Following reviewer's suggestion, we have decided to move Fig 6e-f to the Extended Data Figure 13 and modify the part of the text in which we interpret the results into the light of a possible causal relationship between cytokine responses and LB susceptibility (lines 340 to 342). We still consider these results relevant to understand the role of inflammation and the innate response in the progression and LB, as our results suggest that a sufficient innate response may be necessary to achieve better protection, and this may be relevant to researchers studying immunomodulation.

Minor comments

1. On line 172 you state that "... variants present in the loci had enough alternative alleles". This is confusing as it seems to indicate that you were focusing on multi-allelic variants that had multiple different alternative alleles. Based on the reading of the rest of the manuscript, it seems that you are actually referring to either minor allele count or minor allele frequency (MAF). Could you please clarify this in the text?

We thank the reviewer for the comment. Indeed, we were referring to minor allele counts and have now modified the main text and in the figures.

2. On lines 204-205, while discussing the TLR1-6-10 locus, you mention what Bb and P3C are

both both TLR2 ligands. This is confusing as you do not explain in the text that TLR1 forms a complex with TLR2 and this is the reason why these stimuli are relevant. You do illustrate the complex on Fig 4f, but I would also recommend explicitly mentioning this assumption in the text.

We thank the reviewer for the comment. We have now modified the text to explicitly indicate the *Bb* and P3C are TLR1-TLR2 dimer ligands (line 239).

3. The meaning of the dark and light blue lines on Fig 6e-f has not been explained. Similarly, on Fig 6e-f and Supplementary Figure 12 it is unclear what is exactly considered as outcome.

We thank the reviewer for the comment. We have indicated in the legend that the light blue line indicates the estimate using MR-Egger and the dark blue line indicates the estimate using IVW.

REVIEWER COMMENTS

Reviewer #1 (Remarks to the Author):

The authors have answered all my questions satisfactorily.

Reviewer #3 (Remarks to the Author):

The authors have now addressed most of my concerns, and overall, I am happy with the updated version of the manuscript.

However, I still think that the MR analysis (now presented in Extended Data Figure 13) is misleading. In their response, the authors state that: "Now all plots include the origin of coordinates to show that the current MR Egger estimates are passing through the origin, and we can therefore consider them valid."

I really do not understand how the authors can make such a statement. For reference, I have attached the Extended Data Figure 13 to this review. My understanding is that the MR Egger estimate is represented by the dark blue line that obviously DOES NOT pass through the origin (0,0 point), so the authors' claim that it does is clearly false. As a result, the authors have not addressed my initial concern about interpreting such a regression line.

Moreover, after spending considerable time reviewing MR literature after I wrote my initial review, I think that an even larger issue is overdispersion heterogeneity in the data, whereby individual causal effect estimates provided by different genetic variants are markedly different from each other. Overdispersion heterogeneity is known to inflate the significance of the inverse variance weighted MR (see for example <https://arxiv.org/abs/1512.04486> and <https://doi.org/10.1101/2023.07.20.23292958>). MR best practice guidelines recommend using a multiplicative random-effects MR model instead whenever possible (<https://wellcomeopenresearch.org/articles/4-186/v3>). Quoting from that paper:

"The IVW method can be performed using a fixed-effects or a random-effects meta-analysis model. Unless there are very few variants (meaning that heterogeneity between the variant-specific estimates cannot be estimated reliably) or all variants are taken from the same gene region, we recommend using a multiplicative random-effects model as the default option for the IVW method. If there is no more heterogeneity between the ratio estimates for the individual variants than would be expected by chance alone, then the random-effect analysis is equivalent to the fixed-effect analysis, and there is no loss of precision in making the weaker random-effects assumption. However, if there is excess heterogeneity, then the fixed-effect analysis is inappropriate, as its confidence intervals are misleadingly narrow. A multiplicative random-effects model is preferred to the additive random-effects model that is more common in the meta-analysis literature as it does not change the relative weighting of the variant-specific estimates³⁴. In contrast, an additive random-effects model upweights outlying estimates, which are more likely to represent pleiotropic variants. The multiplicative random-effects IVW method provides valid causal estimates under the assumption of balanced pleiotropy; that is, pleiotropic effects on the outcome are equally likely to be positive as negative³⁴."

Can you please also re-calculate your MR estimates using the multiplicative random-effects model? It is straightforward to do using the `mr_ivw` function from the MendelianRandomization R package. I would also recommend using delta weights to account for the uncertainty of the variant effects on the exposure: `mr_ivw(x, model = "random", weights = "delta")`.

REVIEWER COMMENTS

Reviewer #1 (Remarks to the Author):

The authors have answered all my questions satisfactorily.

We thank the reviewer for their remarks.

Reviewer #3 (Remarks to the Author):

The authors have now addressed most of my concerns, and overall, I am happy with the updated version of the manuscript.

We thank the reviewer for their thorough review of our work and attention to the methods. Their feedback has greatly improved the manuscript.

However, I still think that the MR analysis (now presented in Extended Data Figure 12) is misleading. In their response, the authors state that: "Now all plots include the origin of coordinates to show that the current MR Egger estimates are passing through the origin, and we can therefore consider them valid."

I really do not understand how the authors can make such a statement. For reference, I have attached the Extended Data Figure 12 to this review. My understanding is that the MR Egger estimate is represented by the dark blue line that obviously DOES NOT pass through the origin (0,0 point), so the authors' claim that it does is clearly false. As a result, the authors have not addressed my initial concern about interpreting such a regression line.

We thank the reviewer for their comments.

Regarding the statement of the MR Egger estimate passing through the origin, our initial statement was driven by a mistake, where we interpreted the clear blue line as the MR Egger estimate. The dark one should have been interpreted. We apologize for this mistake and the confusion.

Regarding the original remark by the reviewer, indicating that the MR Egger estimates seem implausible because they pass far from the origin, we provide our new interpretation here:

As the MR Egger result does not go through the origin, according to Burgess et al., 2017¹, the causal estimate provided by MR Egger may be biased. Instead, the results presented now include the estimate from the multiplicative random-effects IVW estimate, as the reviewer suggested, and the MR Egger intercept is used as a sensitivity analysis to test for pleiotropy, rather than for causal estimation. The detailed results can be found below.

The manuscript has been revised accordingly in line 559:

"For multi-snp MR, SNPs with p-value of associations below 10^{-5} were clumped at $R^2 < 0.001$ and used as instruments. Both inverse-variance weighted meta-analysis (multiplicative random-effects) and MR Egger were used for the meta-analysis of the clumped SNPs. This

was performed using the `mr` function from the `TwoSampleMR` package³⁹. Since the MR Egger estimate for the intercept was not zero, this method was not used for causal estimation. IVW was the only method used for causal estimation.”

Moreover, after spending considerable time reviewing MR literature after I wrote my initial review, I think that an even larger issue is overdispersion heterogeneity in the data, whereby individual causal effect estimates provided by different genetic variants are markedly different from each other. Overdispersion heterogeneity is known to inflate the significance of the inverse variance weighted MR (see for example <https://arxiv.org/abs/1512.04486> and <https://doi.org/10.1101/2023.07.20.23292958>). MR best practice guidelines recommend using a multiplicative random-effects MR model instead whenever possible (<https://wellcomeopenresearch.org/articles/4-186/v3>). Quoting from that paper:

"The IVW method can be performed using a fixed-effects or a random-effects meta-analysis model. Unless there are very few variants (meaning that heterogeneity between the variant-specific estimates cannot be estimated reliably) or all variants are taken from the same gene region, we recommend using a multiplicative random-effects model as the default option for the IVW method. If there is no more heterogeneity between the ratio estimates for the individual variants than would be expected by chance alone, then the random-effect analysis is equivalent to the fixed-effect analysis, and there is no loss of precision in making the weaker random-effects assumption. However, if there is excess heterogeneity, then the fixed-effect analysis is inappropriate, as its confidence intervals are misleadingly narrow. A multiplicative random-effects model is preferred to the additive random-effects model that is more common in the meta-analysis literature as it does not change the relative weighting of the variant-specific estimates³⁴. In contrast, an additive random-effects model upweights outlying estimates, which are more likely to represent pleiotropic variants. The multiplicative random-effects IVW method provides valid causal estimates under the assumption of balanced pleiotropy; that is, pleiotropic effects on the outcome are equally likely to be positive as negative³⁴."

Can you please also re-calculate your MR estimates using the multiplicative random-effects model? It is straightforward to do using the `mr_ivw` function from the `MendelianRandomization` R package. I would also recommend using delta weights to account for the uncertainty of the variant effects on the exposure: `mr_ivw(x, model = "random", weights = "delta")`.

We thank the reviewer for reviewing the existing MR guidelines and providing a useful set of resources. The method suggested by the reviewer is useful as it follows the latest guidelines and would help us correct overdispersion heterogeneity in the data. We would like to note that the method we initially used for MR, the `mr` function from the `TwoSampleMR` package, already uses a multiplicative random-effects (MRE) model. Methodologically, the package also provides an alternative method that corrects for under-dispersion heterogeneity. To examine any possible differences that would arise after this correction, we applied the `mr_ivw_mre` function from the same package. The results show the exact same estimates for MRE IVW with and without under-dispersion correction (See revised Extended Data Figure 12, copied below the text).

Due to the relevant points brought up by the reviewer, and following the latest guidelines provided by Burgess et al. 2023², we have modified the revised Extended Data Figure 12 to include the IVW estimate with and without under-dispersion correction and have provided new sensitivity analyses in Extended Data Table 9.

The Guideline provided by the reviewer² states:

“We recommend the IVW method with multiplicative random effects as the primary analysis method for use with summarized data, because it is the most efficient analysis method with valid instrumental variables, and it accounts for heterogeneity in the variant-specific causal estimates. If a causal effect is detected using this method, then investigators should proceed to perform sensitivity analyses (Section 6 and Section 7) to assess the robustness of their finding to the assumption of balanced pleiotropy.”

Following this, we now indicate the estimates and p-values of the IVW method in Extended Data Figure 12 and Extended data Table 9. MRE IVW is robust to heterogeneity.

Nevertheless, to account for possible heterogeneity in the estimation, we have performed a Cochran’s Q test, which suggests a possible excess in heterogeneity in our data (P = 0.06, P = 0.04 for IL-6 and IL-10 respectively).

Since pleiotropy is not controlled for in IVW, we further estimated pleiotropy by estimating the intercept using MR-Egger, which suggests an absence of pleiotropy (P = 0.32, P = 0.06 for IL-6 and IL-10 respectively).

The sensitivity analysis in terms of heterogeneity and pleiotropy are now added in the methods, from line 565 “Cochran’s Q test was applied to test for possible heterogeneity in the IVW estimates. MR Egger’s intercept was used to inspect pleiotropy in the analysis. Both sensitivity measures are listed in Extended Data Table 9”. The text in the Discussion (line 407) has now been revised to “We tested the effect of cytokine production capacity on LB susceptibility by applying Mendelian Randomisation methods. Our results showed a suggestive causal link between higher IL-6 and IL-10 production and protection against LB. However, the results should be interpreted with caution, as we used a suggestive significance threshold (nominal P < 0.05) and the sensitivity tests indicated a probable case of pleiotropy (PIL-6 = 0.32, PIL-10 = 0.06; Extended Data Table 9) and excess in heterogeneity (PIL-6 = 0.06, PIL-10 = 0.04; Extended Data Table 9).”

Overall, these results highlight a potential effect of cytokine production capacity in response to stimulation on Lyme borreliosis susceptibility: IL-10 responses to *Borrelia* at baseline and IL-6 responses at the 6 weeks timepoint. **Immunologically**, the cytokines and the timepoints could point at the determinant effect of higher anti-inflammatory responses (IL-10) in the acute phase of the infection and higher pro-inflammatory responses thereafter (IL-6) as a possible mechanism to achieve a better protection against developing a severe infection.

Given the relevance of these findings for researchers studying the immunological determinants of Lyme Borreliosis susceptibility and the possible **therapeutic targets** that this offers for immunomodulation, we consider necessary to add the data to the revised manuscript. Furthermore, we offer to the reader a range of methods and sensitivity analyses, following the latest guidelines that allow for an informed interpretation of the results.

1. Burgess S, Thompson SG. Interpreting findings from Mendelian randomization using the MR-Egger method *Eur J Epidemiol.* 2017;32(5):377-389. doi:10.1007/s10654-017-0255-x
2. Burgess S, Davey Smith G, Davies NM *et al.* Guidelines for performing Mendelian randomization investigations: update for summer 2023 [version 3; peer review: 2 approved]. Wellcome Open Res 2023, 4:186 (<https://doi.org/10.12688/wellcomeopenres.15555.3>)

The updated **Extended Data Figure 12** is shown below:

Extended Data Figure 12. Mendelian randomization of cytokine production capacity upon *B. burgdorferi* stimulation in PBMCs. Exposures: a, IL-6 response upon *B. burgdorferi* stimulation in PBMCs after antibiotic treatment. b, IL-10 in *B. burgdorferi* stimulated PBMCs at baseline. Dark blue line, multiplicative random effects inverse variance weighted meta-analysis.

IL-6 production upon *Bb* stimulation of PBMCs at the 6 weeks timepoint

IL-10 production upon *Bb* stimulation of PBMCs at the baseline

REVIEWERS' COMMENTS

Reviewer #3 (Remarks to the Author):

The authors have now addressed all of my concerns. I really appreciate the revised MR analysis and more nuanced discussion around it.